# CENTRALITY GRAPH SHIFT OPERATORS FOR GRAPH NEURAL NETWORKS

## ABSTRACT

Graph Shift Operators (GSOs), such as the adjacency and graph Laplacian matrices, play a fundamental role in graph theory and graph representation learning. Traditional GSOs are typically constructed by normalizing the adjacency matrix by the degree matrix, a local centrality metric. In this work, we instead propose and study Centrality GSOs (CGSOs), which normalize adjacency matrices by global centrality metrics such as the PageRank, $k$-core or count of fixed length walks. We study spectral properties of the CGSOs, allowing us to get an understanding of their action on graph signals. We confirm this understanding by defining and running the spectral clustering algorithm based on different CGSOs on several synthetic and real-world datasets. We furthermore outline how our CGSO can act as the message passing operator in any Graph Neural Network and in particular demonstrate strong performance of a variant of the Graph Convolutional Network and Graph Attention Network using our CGSOs on several real-world benchmark datasets.

## 1 INTRODUCTION

We propose and study a new family of operators defined on graphs that we call Centrality Graph Shift Operators (CGSOs). To insert these into the rich history of matrices representing graphs and centrality metrics, the two concepts married in CGSOs, we begin by recalling major advances in these two topics in turn (readers interested purely in recent developments in Graph Representation Learning and Graph Neural Networks are recommended to begin reading in Paragraph 3 of this section). The study of graph theory and with it the use of matrices to represent graphs have a long-standing history. Graph theory is often said to have its origins in 1736 when Leonard Euler posed and solved the Königsberg bridge problem (Euler, 1736). His solution did not involve any matrix calculus. In fact, it seems that the first matrix defined to represent graph structures is the *incidence matrix* defined by Henri Poincaré in 1900 (Poincaré, 1900). It is difficult to pinpoint the first definition of *adjacency matrices*, but by 1936 when the first book on the topic of graph theory was published by Dénes König adjacency matrices had certainly been defined and began to be used to solve graph theoretic problems (König, 1936). Two seemingly concurrent works in 1973 defined an additional matrix structure to represent graphs that later became known as the *unnormalized graph Laplacian* (Donath & Hoffman, 1973; Fiedler, 1973). Then, it was Fan Chung in her book "Spectral Graph Theory" published in 1997 who extensively characterized the spectral properties of *normalized Laplacians* (Chung, 1997). In the emerging field of Graph Signal Processing (GSP) (Sandryhaila & Moura, 2013; Ortega et al., 2018) these different graph representation matrices were all defined to belong to a more general family of operators defined on graphs, the *Graph Shift Operators (GSOs)*. GSOs currently play a crucial role in graph representation learning research, since the choice of GSO, used to represent a graph structure, corresponds to the choice of message passing function in the currently much-used Graph Neural Network (GNN) models.

In parallel to advances in graph representation via matrices, centrality metrics have proved to be insightful in the study of graphs. Chief among them is the success of the PageRank centrality criterion revealing the significance of certain webpages (Brin & Page, 1998) and playing a role in the formation of what is now one of the largest companies worldwide. But also an even older metric, the $k$-core centrality (Seidman, 1983; Malliaros et al., 2020), as well as the degree centrality, closeness centrality, and betweenness centrality, have proven to be impactful in revealing key structural properties of graphs (Freeman, 1977; Zhang & Luo, 2017).

A commonality of the most frequently used GSOs is their property to encode purely local information in the graph, with the adjacency matrix encoding neighborhoods in the graph and the graph Laplacians relying on the node degree, a local centrality metric, to normalize the adjacency matrix. In this work, we study a novel class of GSOs, the Centrality GSOs (CGSOs) that arise from the normalization of the adjacency matrix by centrality metrics such as the PageRank, $k$-core and the count of fixed length walks emanating from a given node. Our CGSOs introduce global information into the graph representation without altering the connectivity pattern encoded in the original GSO and therefore, maintain the sparsity of the adjacency matrix. We provide several theorems characterizing the spectral properties of our CGSOs. We confirm the intuition gained from our theoretical study by running the spectral clustering algorithm on the basis of our CGSOs on 1) synthetic graphs that are generated from a stochastic blockmodel in which each block is sampled from the Barrabasi-Albert model and 2) the real-world Cora graph in which we aim to recover the partition provided by the $k$-core number of each node. We will furthermore describe how our CGSOs can be inserted as the message passing operator into any GNN and observe strong performance of the resulting GNNs on real-world benchmark datasets.

In particular, our contributions can be summarized as follows,

(i) we define Centrality GSOs, a novel class of GSOs based on the normalization of the adjacency matrix with different centrality metrics, such as the degree, PageRank score, $k$-core number, and the count of walks of a fixed length,

(ii) we conduct a comprehensive spectral analysis to unveil the fundamental properties of the CGSOs. Our gained understanding of the benefits of CGSOs is confirmed by running the spectral clustering algorithm using our CGSOs on synthetic and real-world graphs,

(iii) we incorporate the proposed CGSOs within GNNs and evaluate performance of a Graph Convolutional Network and Graph Attention Network v2 with a CGSO message passing operator on several real-world datasets.

## 2 BACKGROUND AND RELATED WORK

We begin by giving a rigorous introduction to GSOs and GNNs.

### 2.1 GRAPH SHIFT OPERATORS

Graph Shift Operators (GSOs) play a pivotal role in the analysis of graph-structured data. Degree-normalized GSOs, such as the Random-walk Normalised Laplacian (Modell & Rubin-Delanchy, 2021), have been widely employed in spectral analysis and signal processing on graphs. These GSOs have many properties allowing great insight in the connectivity of nodes. However, to the best of our knowledge there has been no studies in which the choice of the degree centrality, a local centrality metric, was compared to GSOs in which global centriality metrics are used instead.

We consider graphs $G = (\mathcal{V}, \mathcal{E})$ where $\mathcal{V} = \{1, \dots, N\}$ is the set of nodes, and $\mathcal{E} \subset \mathcal{V} \times \mathcal{V}$ is the set of edges. The adjacency matrix is one of the standard graph representation matrices considered in our work. Formally, a graph can be represented by an adjacency matrix $\mathbf{A} = [a_{ij}] \in \mathbb{R}^{N \times N}$ where $a_{ij} = 1$ if $(i, j) \in \mathcal{E}$ and $a_{ij} = 0$ otherwise. Analyzing the spectrum of the adjacency matrix provides information about the basic topological properties of the underlying graphs (Cvetkovic et al., 1980). For example, the largest eigenvalue of $\mathbf{A}$ is an upper bound of the average degree, a lower bound on the largest degree (Cvetković et al., 2009; Sarkar & Jalan, 2018) and its multiplicity indicates whether the represented graph is connected (Stanić, 2015). Another example is the fact that the adjacency spectrum of bipartite graphs is symmetric around $0$ (Stanić, 2015).

In addition to the adjacency matrix, there are alternative graph representations that provide deep insights into the topology of the underlying graph. One often-used representation is the symmetrically normalized Laplacian matrix defined by $\mathbf{L}_{sym} = \mathbf{I} - \mathbf{D}^{-1/2} \mathbf{A} \mathbf{D}^{-1/2}$, where $\mathbf{D} \in \mathbb{R}^{N \times N}$ is the degree matrix, i.e., a diagonal matrix defined as $\mathbf{D}_{ii} = \sum_{i=1}^{N} a_{ij}$. The normalized Laplacian plays a fundamental role in spectral graph theory. For example, the celebrated *Cheeger's Inequality* establishes a bound on the edge expansion of a graph via its spectrum (Cheeger, 1970). There are other graph representations with particularly interesting spectral properties, such as the random-walk Normalised Laplacian (Modell & Rubin-Delanchy, 2021) and the Signless Laplacian matrices

(Cvetković & Simić, 2009). All these graph representations belong to the family of *Graph Shift Operators* (GSOs), which we define now in Definition 2.1.

**Definition 2.1.** Given an arbitrary graph $G = (\mathcal{V}, \mathcal{E})$, a *Graph Shift Operator* $\mathbf{S} \in \mathbb{R}^{N \times N}$ is a matrix satisfying $\mathbf{S}_{ij} = 0$ for $i \neq j$ and $(i, j) \notin \mathcal{E}$ (Mateos et al., 2019) and $\mathbf{S}_{ij} \neq 0$ for $i \neq j$ and $(i, j) \in \mathcal{E}$.

In addition to the classical or fixed GSOs, parametrized GSOs can be learned during the optimization process of any model in which they are inserted. These parametrized operators are a fundamental component of many modern GNN architectures and allow the model to adapt and capture complex patterns and relationships in the graph data. For example, the work of *PGSO* (Dasoulas et al., 2021) parametrizes the space of commonly used GSOs leading to a learnable GSO that adapts to the dataset and learning task at hand.

## 2.2 GRAPH NEURAL NETWORKS

Graph Neural Networks (GNNs) are neural networks that operate on graph-structured data that is defined as the combination of a graph $G = (\mathcal{V}, \mathcal{E})$, and a node feature matrix $\mathbf{X} \in \mathbb{R}^{N \times d}$, containing the node feature vector of node $i$ in its $i^{\text{th}}$ row. GNNs are formed by stacking several computational layers, each of which produces a hidden representation for each node in the graph, denoted by $\mathbf{H}^{(\ell)} = [h_v^{(\ell)}]_{v \in \mathcal{V}}$. A GNN layer $\ell$ updates node representations relying on the structure of the graph and the output of the previous layer $\mathbf{H}^{(\ell-1)}$. Conventionally, the node features are used as input to the first layer $\mathbf{H}^0 = \mathbf{X}$. The most popular framework of GNNs is that of Message Passing Neural Networks (Gilmer et al., 2017; Hamilton, 2020), where the computations are split into two main steps:

**Message Passing:** Given a node $v$, this step applies a permutation-invariant function to its neighbors, denoted by $\mathcal{N}(v)$, to generate the aggregated representation,

$$\mathbf{M}^{(\ell+1)} = \Phi(\mathbf{A})\mathbf{H}^{(\ell)}, \tag{1}$$

where $\Phi(\mathbf{A}) : \mathbb{R}^{N \times N} \to \mathbb{R}^{N \times N}$, a function of the adjacency matrix, is the chosen GSO.

**Update:** In this step, we combine the aggregated hidden states with the previous hidden representation of the central node $v$, usually by making use of a learnable function,

$$\mathbf{H}^{(\ell+1)} = \sigma(\mathbf{M}^{(\ell+1)}\mathbf{W}^{(\ell)}), \tag{2}$$

where $\mathbf{W}^{(\ell)} \in \mathbb{R}^{d_{\ell-1}, d_\ell}$ are learnable weight matrices and $d_\ell$ is the dimension of the hidden representation at the $\ell$-th layer.

With the emergence and increasing popularity of GNNs, the importance of GSOs has significantly increased. Numerous GNN architectures, such as notably Graph Convolutional Networks (GCNs), rely on these operators in their message passing step. In the context of GCNs (Kipf & Welling, 2016), the used message passing operator, i.e., the chosen GSO, corresponds to $\Phi(\mathbf{A}) = \mathbf{D}_1^{-1/2}\mathbf{A}\mathbf{D}_1^{-1/2}$, where $\mathbf{D}_1 = \mathbf{D} + \mathbf{I}$ is the degree matrix of the graph corresponding to the adjacency matrix $\mathbf{A}_1 = \mathbf{A} + \mathbf{I}$. For Graph Attention Networks v2 (GATv2) (Brody et al., 2022), (1) becomes $\mathbf{M}^{(\ell+1)} = \Phi(\mathbf{A}_{\text{GATv2}}^{(\ell)})\mathbf{H}^{(\ell)}$, where, in this setting, $\Phi$ corresponds to the identity function and the rows of $\mathbf{A}_{\text{GATv2}}^{(\ell)}$ contain the edge-wise attention coefficients.

As we will present shortly, in this work, we generalize the concept of GSOs to encompass global structural information beyond node degree. The proposed CGSO framework encapsulates several global centrality criteria, demonstrating intriguing spectral properties. We further leverage CGSOs to formulate a new class of message passing operators for GNNs, enhancing model flexibility.

**Global Information in GNNs.** Besides our GNNs, which leverage the CGSO to make global information accessible to any given GNN layer, there exists a plethora of other approaches to achieve this goal. These include for example the PPNP and APPNP (Gasteiger et al., 2019), as well as the PPRGo (Bojchevski et al., 2020) models that use the PageRank centrality to define a completely new graph over which to perform message passing in GNNs. The work of Ramos Vela et al. (2022) extends these models to consider both the PageRank and $k$-core centrality. In addition, there is the AdaGCN (Sun et al., 2019) and the VPN model (Jin et al., 2021) which propose to message pass using powers of the adjacency matrix to incorporate global information and increase the robustness

of GNNs, respectively. Lee et al. (2019) propose the Motif Convolutional Networks, that define motif adjacency matrices and then use these in the message passing scheme. Also the $k$-hop GNNs of Nikolentzos et al. (2020) consider neighbors several hops away from a given central node in the message passing scheme of a single GNN layer to consider more global information in a GNN. Additionally there exists a rich and long-standing literature on spectral GNNs that facilitate global information exchange by explicitly or approximately making use of the spectral decomposition of the GSO chosen to be the GNN's message passing operator (Bruna et al., 2014; Defferrard et al., 2016; Koke & Cremers, 2024). Finally, there is an arm of research investigating graph transformers, where usually the graph structure is only used to provide structural encodings of nodes and the optimal message passing operator is learning using an attention mechanism (Kreuzer et al., 2021; Rampášek et al., 2022; Ma et al., 2023). All these approaches increase the computational complexity of the GNN, whereas our CGSO based GNNs maintain the complexity of the underlying GNN model by preserving the sparsity of the original adjacency matrix.

## 3 CGSO: CENTRALITY GRAPH SHIFT OPERATORS

In this section, we introduce the Centrality GSOs (CGSO), a family of shift operators that incorporate the global position of nodes in a graph. We discuss different instances of CGSOs corresponding to widely used centrality criteria. We further conduct a comprehensive spectral analysis to unveil the fundamental properties of CGSOs, including the eigenvalue structure and the expansion properties, examining how these operators influence information spread across the graph. Then, we leverage CGSOs in the design of flexible GNN architectures.

### 3.1 MATHEMATICAL FORMULATION

For a given node $i \in \mathcal{V}$, let $v(i)$ denote a centrality metric associated with $i$, such as the node degree, $k$-core number, PageRank, or the count of walks of specific length starting from node $i$. The Hilbert space $L^2(G)$ is characterized by the set of functions $\varphi$ defined on $\mathcal{V}$ such that $\sum_{i \in \mathcal{V}} v(i)|\varphi(i)|$ converges, equipped with the inner product: $\langle \varphi_1, \varphi_2 \rangle_G = \sum_{i \in \mathcal{V}} v(i)\varphi_1(i)\bar{\varphi}_2(i)$. The *Markov Averaging Operator* on $L^2(G)$ is defined as the linear map $\mathbf{M}_G : \varphi \mapsto \mathbf{M}_G \varphi$ such that

$$\left(\mathbf{M}_G \varphi\right)(i) = \left(\mathbf{V}^{-1}\mathbf{A}\varphi\right)(i) = \frac{1}{v(i)} \sum_{j \in \mathcal{N}_i} \varphi(j),$$

where $\mathbf{V} = diag(v(1), \ldots, v(N))$ and $\mathcal{N}_i$ is the neighborhood set of node $i$. The form of this Markov Averaging Operator gives rise to the simplest formulation of our CGSOs, which is a left normalization of the adjacency matrix by a diagonal matrix containing node centralities on the diagonal, i.e., $\mathbf{V}^{-1}\mathbf{A}$. Note that the *mean aggregation* operator, as discussed in Xu et al. (2019), represents a specific instance of these CGSOs where the degree corresponds to the chosen centrality metric, namely $\mathbf{V} = \mathbf{D}$. We will further extend the concept of CGSOs in (4) where we extend and parameterize these CGSOs. In this paper, we focus on three global centrality metrics, in addition to the local node degree. We recall the definitions of these global centrality metrics now.

**$k$-core.** The $k$-core number of a node can be determined in the process of the $k$-core decomposition of a graph, which captures how well-connected nodes are within their neighborhood (Malliaros et al., 2020). The process of $k$-core decomposition involves iteratively removing vertices with degree less than $k$ until no such vertices remain. The core number $k$ of a node is then equal to the largest $k$ for which the considered nodde is still present in the graph's $k$-core decomposition. We define $\mathbf{V}_{core} \in \mathbb{R}^{N \times N}$ to be the diagonal matrix indicating the core number of each node, i.e., $\forall i \in \mathcal{V}$, $\mathbf{V}_{core}[i,i] = core(i)$.

**PageRank.** We choose $\mathbf{V}_{PR} \in \mathbb{R}^{N \times N}$ such that, $\forall i \in \mathcal{V}$, $\mathbf{V}_{PR}[i,i] = (1 - PR(i))^{-1}$, where $PR(i)$ corresponds to the PageRank score (Brin & Page, 1998). The PageRank score quantifies the likelihood of a random walk visiting a particular node, serving as a fundamental metric for evaluating node significance in various networks.

**Walk Count.** Here, we consider $\mathbf{V}_{\ell\text{-}walks} \in \mathbb{R}^{N \times N}$, the diagonal matrix indicating the number of walks of length $\ell$ starting from each node $i$, i.e., $\forall i \in \mathcal{V}$, $\mathbf{V}_{\ell\text{-}walks}[i,i] = \left(\mathbf{A}^\ell \mathbb{1}\right)[i]$, where $\mathbb{1} \in \mathbb{R}^N$ is the vector of ones. When $\ell = 2$, $\mathbf{V}_{\ell\text{-}walks}$ corresponds to $\mathbf{W}_{\mathbf{M}_{13}}\mathbb{1} - \mathbf{D}$, where $\mathbf{W}_{\mathbf{M}_{13}}$ the graph operator presented by Benson et al. (2016), which corresponds to the count of open bidirectional

wedges, i.e., the motif $M_{13}$. This motif network captures higher-order structures and gives new insights into the organization of complex systems.

In what follows, we delve into the theoretical properties of Markov Averaging Operators, since all three CGSOs $\mathbf{V}_{core}$, $\mathbf{V}_{PR}$ and $\mathbf{V}_{\ell\text{-walks}}$ are instances of Markov Averaging Operators.

**Proposition 3.1.** *The following properties of operator* $\mathbf{M}_G$ *hold.*

*(1)* $\mathbf{M}_G$ *is self-adjoint.*

*(2)* $\mathbf{M}_G$ *is diagonalizable in an orthonormal basis, its eigenvalues are real numbers, and all eigenvalues have absolute values at most* $\gamma = \min_{i \in \mathcal{V}} \left( \frac{v(i)}{deg(i)} \right)$.

The proof of Proposition 3.1 and all subsequent theoretical results in this section can be found in Appendix J. Hence, we have shown in Proposition 3.1 that all CGSOs have a real set of eigenvalues, which is of real use in practice.

In the now following Proposition 3.2 we provide the mean and standard deviation of the spectrum of $M_G$, i.e., the set of $M_G$'s eigenvalues.

**Proposition 3.2.** *The following properties hold for the spectrum of* $M_G$.

*(1) In a graph* $G = (\mathcal{V}, \mathcal{E})$ *with multiple connected components* $\mathcal{C} \subset \mathcal{V}$, *where each connected component* $\mathcal{C}$ *induces a subgraph of* $G$ *denoted by* $G_{\mathcal{C}}$, *a complete set of eigenvectors of* $\mathbf{M}_G$ *can be constructed from the eigenvectors of the different* $\mathbf{M}_{G_{\mathcal{C}}}$, *where eigenvectors of* $\mathbf{M}_{G_{\mathcal{C}}}$ *are extended to have dimension* $N$ *via the addition of zero entries in all entries corresponding to nodes not in the currently considered component* $\mathcal{C}$.

*(2) The mean* $\mu(\mathbf{M}_G)$ *and standard deviation* $\sigma(\mathbf{M}_G)$ *of* $\mathbf{M}_G$'s *spectrum have the following analytic form*

$$\mu(\mathbf{M}_G) = \frac{1}{n} \sum_{i=1}^{n} \frac{1}{v(i)},$$
$$\sigma(\mathbf{M}_G) = \left[ \left( \frac{1}{n} \sum_{(i,j) \in \mathcal{E}} \frac{1}{v(i)v(j)} \right) - \mu(sp_\phi)^2 \right]^{1/2}.$$

We define the *normalized spectral gap* $\lambda_1(G)$ as the smallest non-zero eigenvalue of $\mathbf{I} - \mathbf{M}_G$. In Proposition 3.4, we link $\lambda_1(G)$ to the expansion properties of the graph. In the literature, we characterize graph expansion via the *expansion* or *Cheeger constant* (Chung, 1997), which measures the minimum ratio between the size of a vertex set and the minimum degree of its vertices, reflecting the graph's connectivity. In our work, we generalize this definition to any centrality metric.

**Definition 3.3.** For a graph $G = (\mathcal{V}, \mathcal{E})$ we define the *centrality-based Cheeger constant* $h_v(G)$ as follows

$$h_v(G) = \min \left\{ \frac{|\partial U|}{|U|_v} \mid U \subset V, |U|_v \leq \frac{1}{2}|\mathcal{V}|_v \right\}, \tag{3}$$

where $|\partial U|$ equals the number of vertices that are connected to a vertex in $U$ but are not in $U$, and $|\cdot|_v : U \subset \mathcal{V} \mapsto \sum_{i \in \mathcal{V}} v(i)$. When the chosen centrality is the degree, $h_v(G)$ corresponds to the classical Cheeger constant.

Definition 3.3 allows us to establish a link between the spectrum of our considered Markov operators, i.e., CGSOs, and the centrality-based Cheeger constant in Proposition 3.4.

**Proposition 3.4.** *Let* $G$ *be a connected, non-empty, finite graph without isolated vertices. We have,*

$$\lambda_1(G) \leq \left( 2N \frac{v_+^2}{v_-} \right) h_v(G),$$

*where we denote* $v_- = \min_{i \in \mathcal{V}} v(i)$ *and* $v_+ = \max_{i \in \mathcal{V}} v(i)$.

## 3.2 CGNN: CENTRALITY GRAPH NEURAL NETWORK

CGSOs, as defined above, normalize the adjacency matrix based on the centrality of the nodes, thereby providing a refined representation of graph connectivity. Here, we leverage CGSOs to

design flexible message passing operators in GNNs. Incorporating CGSOs within GNNs aims to harness structural information, enhancing the model's ability to discern subtle topological patterns for prediction tasks. To achieve this, we integrate these operators, without loss of generality, in Graph Convolutional Networks (GCNs) (Kipf & Welling, 2016) and Graph Attention Networks v2 (GATv2)(Brody et al., 2022). We replace the initial shift operator $\Phi(\mathbf{A})$ in (1), with the proposed CGSOs $\Phi(\mathbf{A}, \mathbf{V})$, incorporating different types centrality operators $\mathbf{V}$ defined in Section 3.1.

It has been shown that the maximum PageRank score converges to zero when the total number of nodes is very high (Cai et al., 2021), which is the case in many real-world dense graph data (Leskovec et al., 2010; Leskovec & Mcauley, 2012). Also, the number of walks is high when the expansion of the graph is high. Thus, training a GNN with the proposed CGSOs can lead to numerical instabilities such as vanishing and exploding gradients. To avoid such issues, we can control the range of the eigenvalues of CGSOs. We particularly consider a learnable parameterized CGSO framework which is a generalization of the work of Dasoulas et al. (2021). This has the further advantage that the CGSOs are fit to the given datasets and learning tasks, which leads to more accurate and higher performing graph representation. The exact formula of the new parametrized CGSO is

$$\Phi(\mathbf{A}, \mathbf{V}) = m_1 \mathbf{V}^{e_1} + m_2 \mathbf{V}^{e_2} \mathbf{A}_a \mathbf{V}^{e_3} + m_3 \mathbf{I}_N, \tag{4}$$

where $\mathbf{A}_a = \mathbf{A} + a\mathbf{I}_N$, and $(m_1, m_2, m_3, e_1, e_2, e_3, a)$ are scalar parameters that are learnable via backpropagation. Here $m_1$ controls the additive centrality normalization of the adjacency matrix. The parameter $e_1$ controls whether the additive centrality normalization is performed with an emphasis on large centrality values (for large positive values of $e_1$) or with an emphasis on small centrality values (for large negative values of $e_1$). Similarly, we have $e_2$ and $e_3$ controlling the emphasis on large or small centralities, as well as whether the multiplicative centrality normalization of the adjacency matrix is performed symmetrically or predominantly as a column or row normalization. The parameter $m_2$ controls the magnitude and sign of the adjacency matrix term; in particular, a negative $m_2$ corresponds to a more Laplacian-like CGSO, while a positive $m_2$ gives rise to a more adjacency-like CGSO. Finally, $a$ determines the weight of the self-loops that are added to the adjacency matrix, and $m_3$ controls a further diagonal regularization term of the CGSO. More details on the experimental setup are provided in Section 5.

In our experiments, we notice the best centrality to vary across datasets, although the walk-based centrality CGSO appears to be frequently outperformed by the $k$-core and PageRank CGSO. More particularly, in some cases e.g. PubMed, it is desirable to use local centrality metrics such as the degree, while for other datasets e.g., Cornell, it's preferable to normalize the adjacency with global centrality metrics. In light of this uncertainty, we can opt for a dynamic, trainable choice of centrality by including both local and global centrality-based CGSO in our CGNN; this can be done by summing the CGSO of the degree matrix with the CGSO of a global centrality metric, e.g., $\mathbf{\Phi} = \mathbf{\Phi}(\mathbf{A}, \mathbf{D}) + \mathbf{\Phi}(\mathbf{A}, \mathbf{V}_{core})$. The parameters $m_1, m_2, m_3$ controlling the magnitude of both the local and global CGSOs are then able to learn the relative importance of the local and the global CGSO. In Section 5, we provide experimental results for GNNs with such combined CGSOs.

**Time Complexity.** We recall that the main complexity of our CGCN model is concentrated around the pre-computation of each centrality score. Computing the degree of all nodes in a graph has a time complexity of $\mathcal{O}(|\mathcal{V}| + |\mathcal{E}|)$, where $|\mathcal{V}|$ is the number of nodes and $|\mathcal{E}|$ is the number of edges in the graph (Cormen et al., 2022). For the PageRank algorithm, each iteration requires one vector-matrix multiplication, which on average requires $\mathcal{O}(|\mathcal{V}|^2)$ time complexity. To compute the core numbers of nodes, we iteratively remove nodes with a degree less than a specified value until all remaining nodes have a degree greater than or equal to that value. This operation can be done with a complexity of $\mathcal{O}(|\mathcal{V}| + |\mathcal{E}|)$. Finally, counting the number of walks of length $\ell$ for all the nodes can be done via matrix multiplication $\mathbf{A}^\ell \mathbb{1}$ where $\mathbb{1} \in \mathbb{R}^N$ is the vector of ones. Since our CGSOs preserve the sparsity pattern of the original adjacency matrix, the complexity of the GNNs in which the CGSOs are inserted is unaltered.

## 4 A SPECTRAL CLUSTERING PERSPECTIVE OF CGSOS

In this section, we analyze CGSOs through the lens of spectral clustering (Von Luxburg, 2007; Ng et al., 2001). Spectral clustering is a powerful technique that relies on the spectrum of GSOs to reveal underlying structures within graphs, providing insights into their connectivity properties.

## 4.1 SPECTRAL CLUSTERING ON STOCHASTIC BLOCK BARABÁSI–ALBERT MODELS

Here, we investigate the behavior of CGSOs in the spectral clustering task on synthetic data. Specifically, we propose a new graph generator that is a trivial combination of the well-known Stochastic Block Models (SBM) (Holland et al., 1983) and Barabási–Albert (BA) models (Albert & Barabasi, 2002), we call this generator the Stochastic Block Barabási–Albert Models (SBBAM). We will now discuss the properties and parameterizations of these two graph generators in turn to then discuss their combination in the SBBAMs.

**SBMs.** Firstly, in SBMs the node set of the graph is partitioned into a set of $K$ disjoint blocks $\mathcal{B}_1, \ldots, \mathcal{B}_K$, where both the number and size of these blocks is a parameter of the model. In SBMs edges are drawn uniformly at random with probability $p_{ij}$ for $i, j \in \{1, \ldots, K\}$ between nodes in blocks $\mathcal{B}_i$ and $\mathcal{B}_j$. Note that this parameterization is often simplified by the following constraints $p_{ij} = q$ if $i = j$ and $p_{ij} = p$ if $i \neq j$. SBMs produce graphs which exhibit cluster structure if $p \neq q$, which makes them a common benchmark for clustering algorithms and subject to extensive theoretical study (Abbe, 2018). Note that SBMs can produce both homophilic graphs if $p < q$ and heterophilic graphs if $q > p$ (Lutzeyer, 2020, Figure 1.2).

**BA.** The second ingredient of our SBBAMs are the Barabási–Albert (BA) models (Albert & Barabasi, 2002). This model generates random scale-free networks using a preferential attachment mechanism, which is why these models are also sometimes referred to as preferential attachment (PA) models. In this PA mechanism we start out with a seed graph and then add nodes to it one-by-one at successive time steps. For each added node $r$ edges are sampled between the added node and nodes existing in the graph, where the probability of connecting to existing nodes is proportional to their degree in the graph. Hence, high degree nodes are more likely to have their degree rise even further than low degree nodes in future time steps of the generation process (an effect, that is some time referred to as 'the rich get richer'). BA models characterize several real-world networks (Barabási & Albert, 1999). A key characteristic of a BA model is their degree distribution. In Lemma 4.1, we prove that the density and connectivity of a BA model strongly depend on and positively correlate with the hyperparameter $r$. Thus, we can generate structurally different BA models by choosing different values of $r$. Lemma 4.1 is proved in Appendix K.

**Lemma 4.1.** *Let $G^{BA}$ be a Barabási–Albert graph of $N$ nodes generated with the hyperparameters $N_0 < N$ the initial number of nodes, $r_0 \leq N_0^2$ the initial number of random edges and $r$ the number of added edges at each time step. Then, the average degree in the network is,*

$$\overline{deg}(G^{BA}) = 2r + 2\frac{r_0}{N} - 2N_0\frac{r}{N},$$

*and thus, as the number of nodes grows, i.e., $N \to \infty$, the average degree becomes $\overline{deg}(G^{BA}) \sim 2r$.*

**SBBAMs.** In our SBBAMs we combine SBMs and BA models, by sampling $K$ BA graphs each of size $|\mathcal{B}_1|, \ldots, |\mathcal{B}_K|$ and with parameters $r_1, \ldots, r_K$. We then randomly draw edges between nodes in different BA graphs, $\mathcal{B}_i$ and $\mathcal{B}_j$, uniformly at random with probability $p_{ij}$ for $i, j \in \{1, \ldots, K\}$. In other words, SBBAMs trivally extend SBMs to graph in which each block is generated using a BA model. This allows us to generate graphs with cluster structure, in which the different clusters exhibit potentially interesting centrality distributions, which will serve as an interesting testbed to explore the clustering obtained from the eigenvectors of our CGSOs.

**Experimental Setting.** To better understand the information contained in the spectral decomposition of our CGSOs we will now generate graphs from our SBBAMs and use the spectral clustering algorithm defined on the basis of our CGSOs to attempt to cluster our generated graphs. In our experimental setting, each block or BA graph has 100 nodes and an individual parameter $r$, specifically, $r_1 = 5$, $r_2 = 10$ and $r_3 = 15$. In addition we set $p_{ij} = 0.1$ for all $i \neq j$ with $i, j \in \{1, 2, 3\}$. Figure 3 in Appendix E gives an example of an adjacency matrix sampled from this model. We observe variations in edge density across different blocks and in particular observe homophilic cluster structure in the third block, while the first block appears to be predominantly heterophilic, a rather challenging and interesting structure.

Figure 4 in Appendix F illustrates the $k$-core distribution of the three individual BA blocks and the combined SBBAM. Notably, the $k$-core distribution distinguishes the three BA graphs, while the nodes in the combined graph exhibit less discernibility by $k$-core.

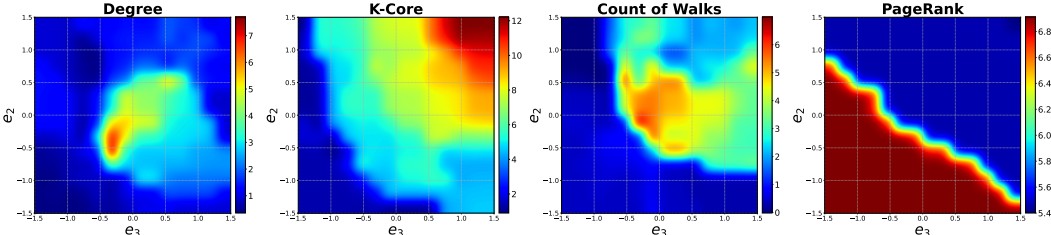

Figure 1: Result for the spectral clustering task on the Cora graph (Sen et al., 2008) with core numbers considered as clusters. We report the values of the Adjusted Mutual Information (AMI) in percentage for different combinations of the exponents $(e_2, e_3)$ in $\mathbf{V}^{e_2}\mathbf{A}\mathbf{V}^{e_3}$.

Following the graph generation, we perform spectral clustering (see Algorithm 1 in Appendix G) using our CGSOs to asses their ability to recover the blocks in our generated SBBAM. Specifically, we utilize the three eigenvectors of $\mathbf{\Phi} = \mathbf{V}^{e_2}\mathbf{A}\mathbf{V}^{e_3}$ corresponding to the three largest eigenvalues of different CGSOs defined in Section 3.1. Working with this particular parametrized form our CGSOs further allows us to study the effect of different centrality normalizations with $e_2, e_3 \in [-1.5, 1.5]$. We repeated each experiment 200 times, and then reported the mean and standard deviation of Adjusted Mutual Information (AMI) and Adjusted Rand Index (ARI) values. For consistency, we used the same 200 generated graphs for all the GSOs and the baselines.

In Figure 1, we report the AMI values using the four centralities. As noticed, while having competitive results between the degree centrality, the PageRank score and the count of walks, we reach the highest AMI values by using the $k$-core centrality metrics. Using the degree centrality, we reach the highest AMI value when both exponent $e_2$ and $e_3$ are negative, while for the $k$-core and the number of walks, we notice a different behavior as the AMI increase when both the exponents $e_2$ and $e_3$ are positive. Thus, we conclude that nodes with higher $k$-core and count of walks are important for this setup, i.e., when the node labels are positively correlated with global centrality metrics such as the $k$-core. We report the ARI values of the same experiment in Appendix H.

## 4.2 Centrality Recovery in Spectral Clustering

In this experiment, we aim to discern the CGSOs' effectiveness in recovering clusters based on centrality within a real-world graph. Using the Cora dataset, we chose core numbers to indicate centrality-based clusters. We aim to assess the capacity of various CGSOs to effectively recover clusters reflective of core numbers. This investigation aims to shed light on their potential utility in capturing centralities and hierarchical structures within intricate graphs.

**Spectral Clustering on Cora.** In this experiment, we consider only the largest connected component of the Cora graph. We use the spectral clustering algorithm on the different CGSOs to recover $K$ clusters, where $K$ is the number of possible core numbers in the graph. We repeat

Table 1: The result of the spectral clustering task on the synthetic graph data. We present the mean and standard values of AMI and ARI in percentage. ① Spectral clustering using the centrality based GSOs, ② Other baselines.

|   | Method | AMI in % | ARI in % |
|---|---|---|---|
| ② | Fast Greedy | 17.27 (4.28) | 19.98 (5.03) |
|   | Louvain | 14.37 (3.34) | 14.82 (3.92) |
|   | Node2Vec | 1.11 (0.92) | 1.17 (0.96) |
|   | Walktrap | 1.39 (1.16) | 1.14 (0.97) |
| ① | CGSO w/ $\mathbf{D}$ | 23.26 (3.36) | 22.95 (3.86) |
|   | CGSO w/ $\mathbf{V}_{core}$ | **35.78 (4.67)** | **33.76 (5.83)** |
|   | CGSO w/ $\mathbf{V}_{\ell\text{-walks}}$ | 23.85 (3.64) | 25.00 (4.18) |
|   | CGSO w/ $\mathbf{V}_{PR}$ | 35.62 (4.90) | 33.19 (6.03) |

each experiment 10 times, and report the average AMI and ARI values. We also compared our CGSOs with the popular Louvain community detection method (Blondel et al., 2008), the node2vec node embedding methods (Grover & Leskovec, 2016) combined with the $k$-means algorithm, the Walktrap algorithm (Pons & Latapy, 2005), and the Fast Greedy Algorithm which also optimizes modularity by greedily adding nodes to communities (Clauset et al., 2004). For the walk count node centrality matrix, we used $\ell = 2$ in all our experiments. We consider the CGSO $\mathbf{\Phi} = \mathbf{V}^{e_2}\mathbf{A}\mathbf{V}^{e_3}$, where we normalize the adjacency matrix with the topological diagonal matrix $\mathbf{V}$ using different exponents $(e_2, e_3)$.

Table 2: Classification accuracy ($\pm$ standard deviation) of the models on different benchmark node classification datasets. The higher the accuracy (in %) the better the model. ① GCN-based models ② Other vanilla GNN baselines ③ CGCN ④ CGATv2. Highlighted are the **first**, second best results. OOM means *Out of memory*.

| | Model | CiteSeer | PubMed | arxiv-year | chameleon | Cornell | deezer-europe | squirrel | Wisconsin |
|---|---|---|---|---|---|---|---|---|---|
| ① | GCN w/ $A$ | 64.95 (0.58) | 77.12 (0.61) | 38.55 (0.71) | 61.03 (1.31) | 57.03 (3.91) | 57.65 (0.84) | 22.38 (6.06) | 54.51 (1.47) |
| | GCN w/ $L$ | 28.11 (0.54) | 43.65 (0.71) | 32.81 (0.29) | 56.97 (0.75) | 54.32 (0.81) | 53.92 (0.59) | 36.20 (0.84) | 60.00 (2.00) |
| | GCN w/ $Q$ | 63.28 (0.80) | 76.57 (0.59) | 33.76 (2.36) | 53.88 (2.35) | 35.41 (2.55) | 56.79 (1.79) | 27.69 (2.21) | 53.33 (0.78) |
| | GCN w/ $L_{rw}$ | 30.18 (0.74) | 59.68 (1.03) | 36.36 (0.24) | 48.77 (0.54) | 61.62 (1.08) | 54.04 (0.44) | 34.27 (0.35) | 65.10 (0.78) |
| | GCN w/ $L_{sym}$ | 29.90 (0.66) | 57.68 (0.45) | 36.49 (0.14) | 50.81 (0.24) | 60.27 (1.24) | 53.30 (0.45) | 35.96 (0.28) | 66.08 (2.16) |
| | GCN w/ $\hat{A}$ | 68.74 (0.82) | 78.45 (0.22) | 42.23 (0.25) | 58.44 (0.26) | 56.22 (1.62) | 60.68 (0.45) | 37.73 (0.33) | 57.45 (0.90) |
| | GCN w/ $H$ | 66.15 (0.55) | 76.45 (0.48) | 41.27 (0.21) | 56.51 (0.47) | 54.86 (1.24) | 59.45 (0.50) | 38.23 (0.47) | 54.31 (0.90) |
| ② | GIN | 66.62 (0.44) | 78.22 (0.52) | 38.27 (3.43) | 61.60 (1.05) | 45.95 (3.42) | | 25.78 (5.12) | 58.82 (1.75) |
| | GAT | 59.84 (3.14) | 71.55 (4.69) | 41.26 (0.30) | 63.60 (1.70) | 49.46 (8.11) | 57.67 (0.74) | 40.37 (2.89) | 55.88 (2.81) |
| | GATv2 | 63.01 (2.97) | 73.96 (2.22) | 41.16 (0.25) | 64.14 (1.53) | 43.78 (4.80) | 56.77 (1.19) | **42.63 (2.61)** | 53.53 (4.12) |
| | PNA | 48.89 (11.15) | 70.83 (6.51) | 32.45 (2.34) | 22.89 (1.09) | 40.54 (0.00) | OOM | OOM | 53.14 (2.55) |
| ③ | CGCN w/ $D$ | 68.35 (0.45) | **78.70 (1.10)** | 45.39 (0.45) | 64.17 (8.10) | 72.43 (13.09) | 58.04 (1.06) | 42.30 (1.34) | 76.86 (7.70) |
| | CGCN w/ $V_{core}$ | 68.40 (0.75) | 77.91 (0.41) | **47.27 (0.31)** | 63.68 (5.00) | 73.78 (12.16) | 60.90 (2.28) | 40.59 (2.21) | 74.90 (6.52) |
| | CGCN w/ $V_{\ell\text{-}walks}$ | 67.31 (0.75) | 77.57 (0.37) | 39.35 (0.49) | **66.21 (2.49)** | 72.70 (3.24) | 59.15 (1.24) | 36.03 (5.81) | 74.90 (4.19) |
| | CGCN w/ $V_{PR}$ | 67.11 (0.56) | 78.17 (4.27) | 47.14 (0.31) | 60.94 (7.00) | 76.22 (16.3) | **63.41 (0.77)** | 32.17 (3.94) | 80.78 (11.7) |
| ④ | CGATv2 w/ $D$ | 68.60 (0.60) | 77.46 (0.51) | 45.09 (0.17) | 58.22 (2.74) | **76.49 (4.37)** | OOM | 35.30 (2.32) | **85.69 (3.17)** |
| | CGATv2 w/ $V_{core}$ | 68.83 (0.66) | 77.99 (0.43) | 44.38 (0.25) | 55.83 (2.28) | 75.95 (3.72) | OOM | 34.17 (1.45) | 85.10 (2.80) |
| | CGATv2 w/ $V_{\ell\text{-}walks}$ | 68.11 (0.91) | 75.43 (0.89) | 46.70 (0.21) | 55.59 (2.57) | 74.32 (5.70) | OOM | 34.25 (2.15) | 83.53 (2.66) |
| | CGATv2 w/ $V_{PR}$ | **68.97 (0.65)** | 78.46 (0.23) | 41.64 (0.18) | 58.82 (1.68) | 74.05 (4.55) | OOM | 38.41 (1.66) | 80.78 (2.45) |

The results of the spectral clustering on this synthetic graph are presented in Table 1. As expected, normalizing the adjacency matrix with $k$-core yields higher AMI and ARI values. This observation indicates an improved discernment of each node's membership in its respective cluster, achieved through the incorporation of global centrality metrics. Our CGSO outperforms well-known community detection techniques, such as the Louvain algorithm, which optimizes the modularity, measuring the density of links inside communities compared to links between communities. However, in our setting, some blocks have fewer inter-edges than intra-edges with other blocks, thus making it difficult for the Louvain algorithm to cluster these nodes using the edge density. This experiment further reinforces the intuition that if different clusters exhibit different centrality distributions then our CGSOs are able to capture this difference better than other clustering alternatives which leads to better clustering performance.

## 5 Experimental Evaluation

We begin by discussing our experimental setup. Further details on the datasets we evaluate on and the training set-up can be found in Appendix A.

**Baselines.** We experiment with two particular instances of our proposed CGNN model, using a GCN and GATv2 as the backbone models, we refer to this instance as CGCN and CGATv2, respectively. We compared the proposed CGCN to GCN with classical GSOs: the adjacency matrix $\mathbf{A}$, Unnormalised Laplacian $\mathbf{L} = \mathbf{D} - \mathbf{A}$, Singless Laplacian $\mathbf{Q} = \mathbf{D} + \mathbf{A}$ (Cvetković & Simić, 2010), Random-walk Normalised Laplacian $\mathbf{L_{rw}} = \mathbf{I} - \mathbf{D}^{-1}\mathbf{A}$, Symmetric Normalised Laplacian $\mathbf{L_{sym}} = \mathbf{I} - \mathbf{D}^{-1/2}\mathbf{A}\mathbf{D}^{-1/2}$, Normalised Adjacency $\hat{\mathbf{A}} = \mathbf{D}^{-1/2}\mathbf{A}\mathbf{D}^{-1/2}$ (Kipf & Welling, 2016) and Mean Aggregation $\mathbf{H} = \mathbf{D}^{-1}\mathbf{A}$ (Xu et al., 2019). We also compare to other standard GNN baselines: Graph Attention Network (GAT) (Veličković et al., 2018), Graph Attention Network v2 (GATv2) (Brody et al., 2022), Graph Isomorphism Network (GIN) (Xu et al., 2019), and Principal Neighbourhood Aggregation (PNA) (Corso et al., 2020).

### 5.1 Experimental Results

We present the performance of our CGCN and CGATv2 in Table 2. The performance of CSGC, i.e. centrality based Simple Graph Convolutional Networks (Wu et al., 2019), in Appendix C. We also incorporated our learnable CGSOs into H2GCN (Zhu et al., 2020) resulting CH2GCN, that go beyond the message passing scheme and which is designed for heterophilic graphs, we detailed the experiment and the results in Appendix I. The results of CGCN, CGATv2, CSGC and the other baselines on additional datasets can be found in Table 7 of Appendix B, and Tables 8 and 9 of Appendix C. It has been observed that, across numerous datasets, CGCN and CGATv2 outperform

Table 3: Classification accuracy ($\pm$ standard deviation) of the models on different benchmark node classification datasets. The higher the accuracy (in %) the better the model.

| Model | CiteSeer | PubMed | arxiv-year | chamelon | Cornell | deezer-europe | squirrel | Wisconsin |
|---|---|---|---|---|---|---|---|---|
| CGCN w/ $\mathbf{D}$ | 68.35 (0.45) | 78.70 (1.10) | 45.39 (0.45) | 64.17 (8.10) | 72.43 (13.09) | 58.04 (1.06) | 42.30 (1.34) | 76.86 (7.70) |
| CGCN w/ $\mathbf{V}_{core}$ | 68.40 (0.75) | 77.91 (0.41) | 47.27 (0.31) | 63.68 (5.00) | 73.78 (12.16) | **60.90 (2.28)** | 40.59 (2.21) | 74.90 (6.52) |
| CGCN w/ $\mathbf{V}_{\ell\text{-walks}}$ | 67.31 (0.75) | 77.57 (0.37) | 39.35 (0.49) | **66.21 (2.49)** | 72.70 (3.24) | 59.15 (1.24) | 36.03 (5.81) | 74.90 (4.19) |
| CGCN w/ $\mathbf{V}_{PR}$ | 67.11 (0.56) | 78.17 (4.27) | 47.14 (0.31) | 60.94 (7.00) | **76.22 (16.3)** | 63.41 (0.77) | 32.17 (3.94) | 80.78 (11.7) |
| CGCN w/ $\mathbf{D} - \mathbf{V}_{core}$ | **69.0 (0.64)** | **78.77 (0.34)** | 48.37 (0.15) | 65.04 (4.37) | 73.24 (6.56) | 59.81 (0.51) | 40.74 (4.77) | 74.51 (3.62) |
| CGCN w/ $\mathbf{D} - \mathbf{V}_{\ell\text{-walks}}$ | 67.99 (0.55) | 78.53 (0.39) | **49.12 (0.41)** | 58.09 (3.78) | 74.32 (2.77) | 59.30 (0.70) | 34.49 (2.66) | **81.37 (3.64)** |
| CGCN w/ $\mathbf{D} - \mathbf{V}_{PR}$ | 68.45 (0.6) | 77.75 (0.55) | 39.63 (1.27) | 64.32 (3.13) | 72.97 (4.98) | 59.28 (0.75) | **42.80 (6.58)** | 74.31 (3.97) |

classical GSOs and vanilla GNNs. Moreover, it is noteworthy that the optimal choice of centrality for CGCN varies depending on the specific dataset. To better understand the choice of each centrality, we displayed the learned weights of CGCN together with some statistics of each dataset in Tables 12, 13, 14 and 15. Several trends are clear: *i)* For all the centrality metrics, the exponent $e_1$ is usually positive for most of the datasets, which indicates that an additive normalization of the GSO with our centralities in-style of the unnormalized Laplacian often leads to optimal graph representation. However, the exponent values $e_2$ and $e_3$ have different behaviors across centrality metrics, e.g., when using the PageRank centrality, the exponents $e_2$ and $e_3$ are almost null for the graph datasets that are strongly homophilous indicating that an unnormalized sum over neighborhoods is optimal. *ii)* When using the PageRank and Count of walks centrality metrics, we notice that the parameter $a$ is always negative for non-homophilous datasets. This is a very interesting finding indicating that a representation with negatively weighted self-loops is advantageous for non-homophilous datasets (an observation that we have not previously seen in the literature). *iii)* For the datasets where the $k$-core centrality performs well (i.e. Cornell, arxiv-year, Penn94, and deezer-europe), we notice that the parameter $m_3$ is very close to zero, i.e, the regularization by adding an identity matrix to the CGSO turns out to be best-ignored in these settings. These findings suggest that the optimal GSO components vary depending on the graph type, highlighting the need for adaptable CGSO approaches rather than relying solely on classical GSOs.

General intuition on the choice of centrality that we can provide relates to the fact that the node degree is a local centrality metric, while the remaining three centralities we consider correspond to global metrics. Therefore, it is apparent that if the learning task only requires local information a degree-based normalization of the GSO is likely beneficial, while global centrality metrics are appropriate if more global information is required. Beyond this statement it seems to be difficult to provide general guidance on the choice of the global centrality metrics. Therefore, including both local and global centrality-based CGSO in the CGNN might be optimal to dynamically distinguish the best type of centrality. We present the results of this experiment in Tables 3 and 10. By combining local and global centralities in the CGNNs, we usually increase their performance.

## 6 CONCLUSION AND LIMITATIONS

**Conclusion.** In this work, we have proposed CGSOs, a novel class of Graph Shift Operators (GSOs) that can leverage different centrality metrics, such as node degree, PageRank score, core number, and the count of walks of a fixed length. Furthermore, we have modified the message-passing steps of Graph Neural Networks (GNNs) to integrate these CGSOs, giving rise to a novel model class the CGNNs. Experimental results comparing our CGNN models to existing vanilla GNNs show the superior performance of CGNN on many real-world datasets. These experiments furthermore allowed us to analyse the optimal parameters of our CGSO, which led to new and interesting insight such as for example an apparent benefit of negatively weighted self-loops for non-homophilous graphs. To further understand the cases where each centrality is beneficial, we conducted additional experiments focused on spectral clustering using two distinct types of synthetic graphs. Through these experiments, we identified instances where CGSOs outperformed conventional GSOs.

**Limitations.** As for the limitations of our approach, rather trivially, the inclusion of centrality metrics is only beneficial if centrality metrics are related to our currently performed learning task on a given dataset. In our experiments, we often observe this case. However, this will not hold for all learning tasks and datasets. In future work, we aim to test the performance of CGNNs on other graph tasks, e.g., link prediction, and to more carefully analyse the learned CGSO parameters. We also aim to extend our theoretical understanding of the CGSOs via further spectral study.

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

## A DATASETS AND IMPLEMENTATION DETAILS

### A.1 STATISTICS OF THE NODE CLASSIFICATION DATASETS

We use ten widely used datasets in the GNN literature. In particular, we run experiments on the node classification task using the citation networks Cora, CiteSeer, and PubMed (Sen et al., 2008), the co-authorship networks CS and Physiscs (Shchur et al., 2018), the citation network between Computer Science arXiv papers OGBN-Arxiv (Hu et al., 2020), the Amazon Computers and Amazon Photo networks (Shchur et al., 2018), the non-homophilous datasets Penn94 (Traud et al., 2012), genius (Lim & Benson, 2021), deezer-europe (Rozemberczki & Sarkar, 2020) and arxiv-year (Hu et al., 2020), and the disassortative datasets Chameleon, Squirrel (Rozemberczki et al., 2021), and Cornell, Texas, Wisconsin from the WebKB dataset (Lim et al., 2021). Characteristics and information about the datasets utilized in the node classification part of the study are presented in Table 4.

Table 4: Statistics of the node classification datasets used in our experiments.

| DATASET | #FEATURES | #NODES | #EDGES | #CLASSES | EDGE HOMOPHILY |
|---|---|---|---|---|---|
| CORA | 1,433 | 2,708 | 5,208 | 7 | 0.809 |
| CITESEER | 3,703 | 3,327 | 4,552 | 6 | 0.735 |
| PUBMED | 500 | 19,717 | 44,338 | 3 | 0.802 |
| CS | 6,805 | 18,333 | 81,894 | 15 | 0.808 |
| ARXIV-YEAR | 128 | 169,343 | 1,157,799 | 5 | 0.218 |
| CHAMELEON | 2,325 | 2,277 | 62,792 | 5 | 0.231 |
| CORNELL | 1,703 | 183 | 557 | 5 | 0.132 |
| DEEZER-EUROPE | 31,241 | 28,281 | 185,504 | 2 | 0.525 |
| SQUIRREL | 2,089 | 5,201 | 396,846 | 5 | 0.222 |
| WISCONSIN | 1,703 | 251 | 916 | 5 | 0.206 |
| TEXAS | 1,703 | 183 | 574 | 5 | 0.111 |
| PHOTO | 745 | 7,650 | 238,162 | 8 | 0.827 |
| OGBN-ARXIV | 128 | 169,343 | 2,315,598 | 40 | 0.654 |
| COMPUTERS | 767 | 13752 | 491,722 | 10 | 0.777 |
| PHYSICS | 8,415 | 34,493 | 495,924 | 5 | 0.931 |
| PENN94 | 4,814 | 41,554 | 2,724,458 | 3 | 0.470 |

### A.2 IMPLEMENTATION DETAILS

We train all the models using the Adam optimizer (Kingma & Ba, 2014). To account for the impact of random initialization, each experiment was repeated 10 times, and the mean and standard deviation of the results were reported. The experiments have been run on both a NVIDIA A100 GPU and a RTX A6000 GPU.

**Training of our CGNN.** We train our model using the Adam optimizer (Kingma & Ba, 2014), with a weight decay on the parameters of $5 \times 10^{-4}$, an initial learning rate of $0.005$ for the exponential parameters and an initial learning rate of $0.01$ for all other model parameters. We repeated the training 10 times to test the stability of the model. We tested 7 initialization of the weights $(m_1, m_2, m_3, e_1, e_2, e_3, a)$. These initializations are reported in Table 5 in Appendix A, and correspond to classical GSOs when the chosen centrality is the degree. For the Cora, CiteSeer, and Pubmed datasets, we used the provided train/validation/test splits. For the remaining datasets, we followed the framework of Lim et al. (2021); Rozemberczki et al. (2021).

### A.3 WEIGHTS INITIALIZATION

In this part, we present the different initializations of CGSO. When the chosen centrality is the degree, i.e. $\mathbf{V} = \mathbf{D}$, the initializations corresponds to popular classical GSO (Dasoulas et al., 2021).

### A.4 HYPERPARAMETER CONFIGURATIONS

For a more balanced comparison, however, we use the same training procedure for all the models. The hyperparameters in each dataset where performed using a Grid search on the classical GCN (i.e. with the GSO : Normalised adjacency) over the following search space:

Table 5: Differenet initialization of the weights $(m_1, m_2, m_3, e_1, e_2, e_3, a)$.

| Initialization of $(\mathbf{m_1}, \mathbf{m_2}, \mathbf{m_3}, \mathbf{e_1}, \mathbf{e_2}, \mathbf{e_3}, \mathbf{a})$ | Corresponding GSO | Description when $\mathbf{V} = \mathbf{D}$ |
|---|---|---|
| $(0, 1, 0, 0, 0, 0, 0)$ | $\mathbf{A}(\mathbf{V}) = \mathbf{A}$ | Adjacency matrix |
| $(1, -1, 0, 1, 0, 0, 0)$ | $\mathbf{L}(\mathbf{V}) = \mathbf{V} - \mathbf{A}$ | Unnormalised Laplacian matrix |
| $(1, 1, 0, 1, 0, 0, 0)$ | $\mathbf{Q}(\mathbf{V}) = \mathbf{V} + \mathbf{A}$ | Signless Laplacian matrix |
| $(0, -1, 1, 0, -1, 0, 0)$ | $\mathbf{L_{rw}}(\mathbf{V}) = \mathbf{I} - \mathbf{V}^{-1}\mathbf{A}$ | Random-walk Normalised Laplacian |
| $(0, -1, 1, 0, -1/2, -1/2, 0)$ | $\mathbf{L_{sym}}(\mathbf{V}) = \mathbf{I} - \mathbf{V}^{-1/2}\mathbf{A}V^{-1/2}$ | Symmetric Normalised Laplacian |
| $(0, 1, 0, 0, -1/2, -1/2, 1)$ | $\hat{\mathbf{A}}(\mathbf{V}) = \mathbf{V}^{-1/2}\mathbf{A}_1\mathbf{V}^{-1/2}$ | Normalised Adjacency matrix |
| $(0, 1, 0, 0, -1, 0, 0)$ | $\mathbf{H}(\mathbf{V}) = \mathbf{V}^{-1}\mathbf{A}$ | Mean Aggregation Operator |

- Hidden size : $[16, 32, 64, 128, 256, 512]$,
- Learning rate : $[0.1, 0.01, 0.001]$,
- Dropout probability: $[0.2, 0.3, 0.4, 0.5, 0.6, 0.7, 0.8]$.

The number of layers was fixed to 2. The optimal hyperparameters can be found in Table 6.

Table 6: Hyperparameters used in our experiments.

| DATASET | HIDDEN SIZE | LEARNING RATE | DROPOUT PROBABILITY |
|---|---|---|---|
| CORA | 64 | 0.01 | 0.8 |
| CITESEER | 64 | 0.01 | 0.4 |
| PUBMED | 64 | 0.01 | 0.2 |
| CS | 512 | 0.01 | 0.4 |
| ARXIV-YEAR | 512 | 0.01 | 0.2 |
| CHAMELEON | 512 | 0.01 | 0.2 |
| CORNELL | 512 | 0.01 | 0.2 |
| DEEZER-EUROPE | 512 | 0.01 | 0.2 |
| SQUIRREL | 512 | 0.01 | 0.2 |
| WISCONSIN | 512 | 0.01 | 0.2 |
| TEXAS | 512 | 0.01 | 0.2 |
| PHOTO | 512 | 0.01 | 0.6 |
| OGBN-ARXIV | 512 | 0.01 | 0.5 |
| COMPUTERS | 512 | 0.01 | 0.2 |
| PHYSICS | 512 | 0.01 | 0.4 |
| PENN94 | 64 | 0.01 | 0.2 |

## B  ADDITIONAL RESULTS FOR THE NODE CLASSIFICATION TASK

To further evaluate our *CGCN* and *CGATv2*, we compute its performance on additional datasets. The results of this study are presented in Table 7.

Table 7: Classification accuracy ($\pm$ standard deviation) of the models on different benchmark node classification datasets. The higher the accuracy (in %) the better the model. ① GCN Based models ② Other Vanilla GNN baselines ③ CGCN ④ CGATv2. Highlighted are the **first**,second best results. OOM means *Out of memory*

| | Model | Cora | Texas | Photo | ogbn-ariv | CS | Computers | Physics | Penn94 |
|---|---|---|---|---|---|---|---|---|---|
| | GCN w/ $A$ | 78.61 (0.51) | 63.51 (2.18) | 82.31 (2.61) | 13.23 (6.44) | 87.70 (1.25) | 69.32 (3.64) | 88.92 (1.93) | 52.35 (0.36) |
| | GCN w/ $\mathbf{L}$ | 31.57 (0.41) | 84.32 (2.65) | 27.42 (6.23) | 10.91 (1.49) | 23.75 (3.22) | 26.27 (3.89) | 35.31 (3.71) | 65.31 (0.59) |
| | GCN w/ $\mathbf{Q}$ | 77.32 (0.50) | 60.54 (1.32) | 77.06 (6.73) | 10.50 (1.97) | 89.42 (1.31) | 47.72 (18.37) | 90.69 (2.13) | 53.46 (2.16) |
| ① | GCN w/ $\mathbf{L_{rw}}$ | 26.59 (1.11) | 78.38 (2.09) | 24.60 (4.21) | 8.07 (0.07) | 26.34 (4.09) | 13.76 (3.96) | 28.19 (3.75) | 69.82 (0.44) |
| | GCN w/ $\mathbf{L_{sym}}$ | 26.79 (0.50) | 71.35 (1.32) | 22.82 (2.67) | 20.18 (0.24) | 24.39 (1.96) | 16.06 (5.19) | 30.94 (3.11) | 70.57 (0.30) |
| | GCN w/ $\hat{\mathbf{A}}$ | 80.84 (0.40) | 60.81 (1.81) | 78.94 (1.65) | 65.80 (0.14) | 91.52 (0.75) | 68.91 (3.00) | 93.72 (0.80) | 74.60 (0.42) |
| | GCN w/ $\mathbf{H}$ | 80.15 (0.37) | 59.46 (0.00) | 73.95 (4.75) | 63.34 (0.15) | 90.98 (1.84) | 62.01 (4.36) | 92.16 (1.12) | 71.78 (0.47) |
| | GIN | 79.06 (0.47) | 57.03 (1.89) | 83.00 (2.52) | 9.30 (6.42) | 89.53 (1.20) | 55.89 (13.45) | 89.15 (2.44) | OOM |
| ② | GAT | 77.73 (1.83) | 52.16 (6.74) | 71.56 (3.48) | 67.36 (0.13) | 67.67 (3.96) | 59.73 (3.59) | 80.91 (4.48) | 73.85 (1.38) |
| | GATv2 | 74.53 (2.48) | 48.11 (3.78) | 73.49 (2.49) | 68.14 (0.07) | 70.13 (4.92) | 58.18 (4.76) | 83.28 (3.68) | 75.54 (2.54) |
| | PNA | 56.67 (10.53) | 63.51 (4.05) | 16.75 (5.59) | OOM | OOM | 13.62 (6.39) | OOM | OOM |
| | CGCN w/ $\mathbf{D}$ | 79.45 (0.58) | 81.89 (9.38) | 88.78 (1.74) | 69.09 (0.21) | 91.28 (1.29) | 79.26 (1.87) | 92.51 (1.16) | 73.06 (0.34) |
| ③ | CGCN w/ $\mathbf{V}_{core}$ | 79.80 (0.43) | 77.84 (5.51) | 88.53 (1.40) | 65.54 (0.57) | 91.37 (1.18) | 77.35 (2.67) | 91.98 (1.49) | 78.11 (3.74) |
| | CGCN w/ $\mathbf{V}_{\ell\text{-}walks}$ | 79.52 (0.35) | 78.11 (5.82) | 83.72 (2.03) | 22.54 (8.22) | 89.87 (1.20) | 68.56 (3.39) | 89.84 (2.74) | 68.44 (0.37) |
| | CGCN w/ $\mathbf{V}_{PR}$ | 79.51 (15.01) | 82.70 (4.95) | 81.28 (6.08) | 68.56 (0.18) | 88.76 (30.68) | 65.54 (6.43) | 89.64 (10.3) | 72.59 (0.84) |
| | CGATv2 w/ $\mathbf{D}$ | 79.07 (0.64) | 82.7 (5.30) | 87.97 (1.77) | 70.09 (0.10) | 91.48 (1.05) | 78.62 (2.35) | 91.32 (1.18) | 72.81 (0.36) |
| ④ | CGATv2 w/ $\mathbf{V}_{core}$ | 79.03 (0.96) | 83.78 (6.62) | 89.72 (1.54) | 69.93 (0.13) | 91.91 (1.06) | 77.31 (3.33) | 91.15 (1.07) | 72.86 (0.41) |
| | CGATv2 w/ $\mathbf{V}_{\ell\text{-}walks}$ | 78.58 (0.58) | 79.73 (4.72) | 88.11 (2.02) | 70.51 (0.24) | 90.73 (1.46) | 79.09 (1.66) | 89.98 (1.37) | 72.79 (0.43) |
| | CGATv2 w/ $\mathbf{V}_{PR}$ | 78.6 (0.38) | 83.78 (4.98) | 88.38 (2.09) | 69.26 (0.12) | 91.77 (1.00) | 74.95 (3.05) | 92.73 (1.44) | 75.16 (0.69) |

## C    SIMPLE GRAPH CONVOLUTIONAL NETWORKS

In Tables 8 and 9, we present the results of our centrality-aware Simple Graph Convolutional Networks *CGSC* of 2 layers. As noticed in most cases, by incorporating our CGSO, we outperform the classical SGC. To also understand the effect of the centrality on the oversmoothing effect, we analyzed the variation of Dirichlet Energy (Zhao et al., 2024) of *CGSC* across different numbers of layers. As noticed, while the centrality has a lower effect on the oversmoothing in the homophilous dataset Cora, we notice a larger impact on the heterophilious dataset Chameleon.

Table 8: Classification accuracy ($\pm$ standard deviation) of the models on different benchmark node classification datasets. The higher the accuracy (in %) the better the model. ① CSGC with nodes centrality, ② SGC. Highlighted are the **best results**.

| | Model | CiteSeer | PubMed | arxiv-year | chamelon | Cornell | deezer-europe | squirrel | Wisconsin |
|---|---|---|---|---|---|---|---|---|---|
| | CSGC w/ $\mathbf{D}$ | 67.70 (0.17) | 77.37 (0.25) | 35.01 (0.16) | 59.10 (1.66) | 72.70 (4.26) | 58.22 (0.47) | 40.12 (1.69) | 75.88 (4.96) |
| ① | CSGC w/ $\mathbf{V}_{core}$ | 66.85 (0.15) | 78.19 (0.12) | 37.71 (0.17) | 63.11 (4.56) | 72.16 (5.55) | 61.29 (0.50) | 38.66 (2.27) | 75.10 (4.12) |
| | CSGC w/ $\mathbf{V}_{\ell\text{-}walks}$ | 67.09 (0.05) | 77.50 (0.18) | 36.67 (0.22) | 45.26 (2.51) | 74.32 (6.07) | 59.69 (0.50) | 27.85 (1.38) | 81.76 (3.73) |
| | CSGC w/ $\mathbf{V}_{PR}$ | 64.91 (0.47) | 76.47 (0.37) | 23.87 (0.51) | 55.18 (3.36) | 69.46 (5.14) | 58.94 (0.48) | 26.73 (2.25) | 75.29 (4.31) |
| ② | SGC | 64.96 (0.10) | 75.72 (0.12) | 26.61 (0.24) | 38.44 (4.41) | 45.41 (5.77) | 62.66 (0.48) | 19.88 (0.79) | 53.53 (8.09) |

Table 9: Classification accuracy ($\pm$ standard deviation) of the models on different benchmark node classification datasets. The higher the accuracy (in %) the better the model. ① CSGC with nodes centrality, ② SGC. Highlighted are the **best results**.

| | Model | Cora | Texas | Photo | ogbn-ariv | CS | Computers | Physics | Penn94 |
|---|---|---|---|---|---|---|---|---|---|
| | CSGC w/ $\mathbf{D}$ | 80.10 (0.11) | 76.76 (3.24) | 89.38 (1.81) | 67.94 (0.06) | 92.29 (1.04) | 79.04 (1.94) | 92.32 (1.2) | 78.84 (4.15) |
| ① | CSGC w/ $\mathbf{V}_{core}$ | 78.80 (0.17) | 77.30 (3.86) | 88.58 (1.68) | 62.54 (0.16) | 91.82 (1.10) | 76.46 (2.29) | 91.71 (1.63) | 76.25 (1.21) |
| | CSGC w/ $\mathbf{V}_{\ell\text{-}walks}$ | 77.32 (0.29) | 80.27 (5.41) | 88.78 (2.69) | 66.41 (0.05) | 91.96 (0.84) | 76.17 (4.92) | 91.71 (1.58) | 73.20 (0.36) |
| | CSGC w/ $\mathbf{V}_{PR}$ | 76.92 (0.39) | 77.30 (4.86) | 84.33 (3.06) | 44.82 (1.16) | 90.24 (0.86) | 61.51 (2.71) | 91.57 (1.70) | 77.24 (0.67) |
| ② | SGC | 78.79 (0.13) | 58.65 (4.20) | 24.0 (11.82) | 60.48 (0.14) | 70.78 (5.47) | 11.34 (11.67) | 91.69 (1.48) | 66.63 (0.62) |

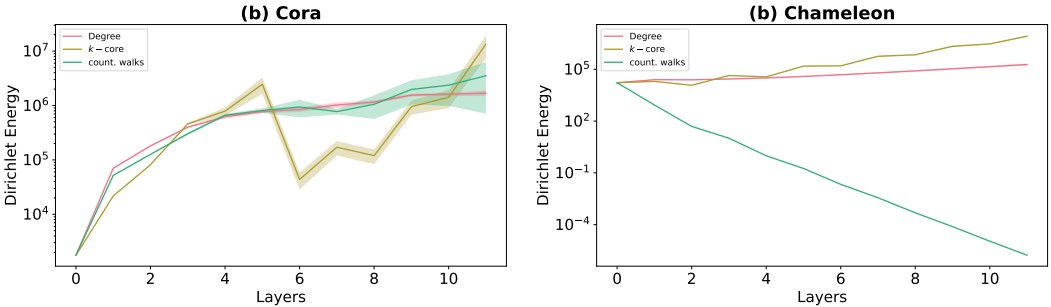

Figure 2: Dirichlet Energy variation with layers in (a) Cora and (b) Chamelon.

## D COMBINING LOCAL AND GLOBAL CENTRALITIES

Table 10: Classification accuracy ($\pm$ standard deviation) of the models on different benchmark node classification datasets. The higher the accuracy (in %) the better the model.

| Model | Cora | Texas | Photo | ogbn-ariv | CS | Computers | Physics | Penn94 |
|---|---|---|---|---|---|---|---|---|
| CGCN w/ $\mathbf{D}$ | 79.45 (0.58) | 81.89 (9.38) | 88.78 (1.74) | 69.09 (0.21) | 91.28 (1.29) | **79.26 (1.87)** | **92.51 (1.16)** | 73.06 (0.34) |
| CGCN w/ $\mathbf{V}_{core}$ | 79.80 (0.43) | 77.84 (5.51) | 88.53 (1.40) | 65.54 (0.57) | 91.37 (1.18) | 77.35 (2.67) | 91.98 (1.49) | 78.11 (3.74) |
| CGCN w/ $\mathbf{V}_{\ell\text{-walks}}$ | 79.52 (0.35) | 78.11 (5.82) | 83.72 (2.03) | 22.54 (8.22) | 89.87 (1.20) | 68.56 (3.39) | 89.84 (2.74) | 68.44 (0.37) |
| CGCN w/ $\mathbf{V}_{PR}$ | 79.51 (15.01) | **82.70 (4.95)** | 81.28 (6.08) | 68.56 (0.18) | 88.76 (30.68) | 65.54 (6.43) | 89.64 (10.3) | 72.59 (0.84) |
| CGCN w/ $\mathbf{D}$ & $\mathbf{V}_{core}$ | **79.88 (0.38)** | 78.92 (4.32) | **89.06 (1.28)** | 67.67 (0.26) | 91.63 (0.95) | 78.41 (1.94) | 91.28 (3.17) | **80.28 (2.93)** |
| CGCN w/ $\mathbf{D}$ & $\mathbf{V}_{\ell\text{-walks}}$ | 79.38 (0.72) | 81.89 (4.69) | 86.78 (2.75) | **69.57 (0.24)** | **91.78 (1.04)** | 78.39 (2.36) | 91.2 (1.56) | 72.5 (0.48) |
| CGCN w/ $\mathbf{D}$ & $\mathbf{V}_{PR}$ | 79.84 (0.4) | 78.11 (2.55) | 82.76 (2.06) | 21.28 (9.89) | 90.04 (0.57) | 65.66 (4.96) | 90.24 (1.86) | 71.03 (5.83) |

## E THE GRAPH STRUCTURE OF THE STOCHASTIC BLOCK BARABÁSI–ALBERT MODELS

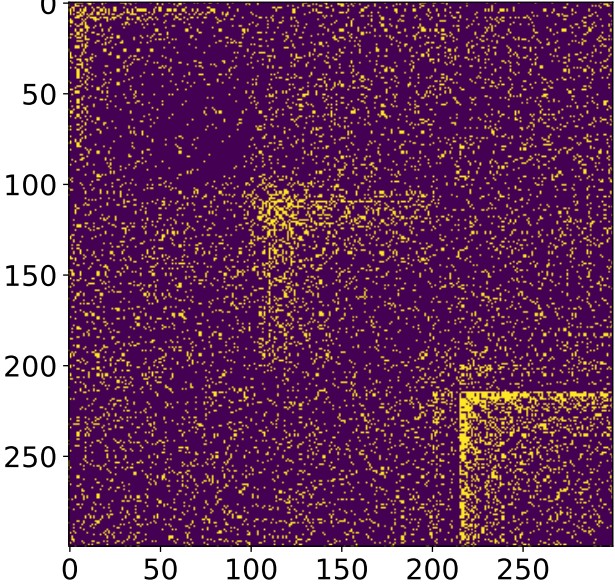

Figure 3: The adjacency matrix of the synthetic graph generated through the combination of three distinct BA models.

# F $k$-CORE DISTRIBUTION IN STOCHASTIC BLOCK BARABÁSI–ALBERT MODELS

In Figure 4, we illustrate the $k$-core distribution of the three individual graphs and the combined graph.

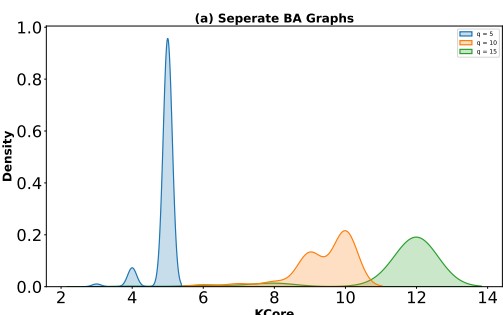 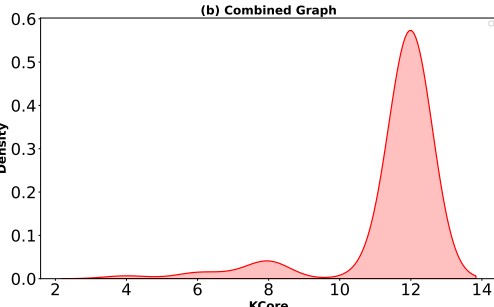

Figure 4: The left figure represents the $k$-core distributions of three different BA models with the hyperparameters $q = 5, 10$ and $15$. The right figure represents the $k$-core distribution of the supra-graph obtained by merging the three BA models.

# G SPECTRAL CLUSTERING ALGORITHM

---

**Algorithm 1:** Spectral Clustering using the Centrality GSOs

---

**Inputs:** Graph $G$, Centrality GSO $\Phi$, Number of clusters to retrieve $C$.

1. Compute the eigenvalues $\{\lambda\}_{i=1}^{n}$ and eigenvectors $\{u\}_{i=1}^{n}$ of $\Phi$;

2. Consider only the eigenvectors $U \in \mathbb{R}^{N \times C}$ corresponding to the $C$ largest eigenvalues;

3. Cluster rows of $U$, corresponding to nodes in the graph, using the $K$-Means algorithm to retrieve a node partition $P$ with $C$ clusters;
$$\mathcal{P} = \text{K-Means}(U, C)$$

**return** $\mathcal{P}$;

---

# H ADDITIONAL RESULTS FOR THE SPECTRAL CLUSTERING TASK

In this section, we report the ARI value of the spectral clustering task described in Section 4.

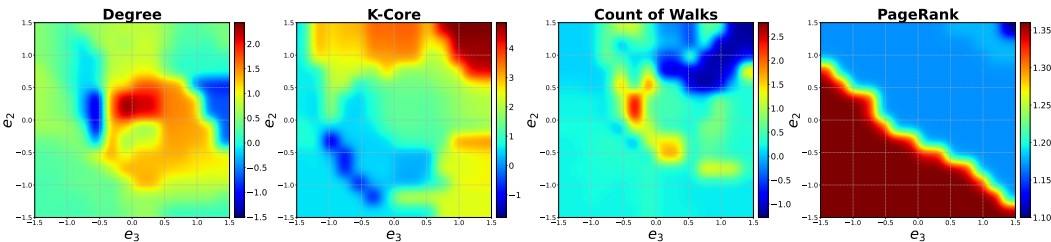

Figure 5: Result for the spectral clustering task on Cora graph with core numbers considered as clusters. We report the values of the Adjusted Rand Information (ARI) in % different combination of the exponents $(e_2, e_3)$ in $\mathbf{V}^{e_2}\mathbf{A}\mathbf{V}^{e_3}$.

# I CGNN WITH HETEROPHILY

In this section, we incorporate our learnable CGSOs into *H2GCN* Zhu et al. (2020), designed for heterophilic graphs. We compared the results of *CH2GCN* and *H2GCN* on datasets with low homophily. We report the results of this experiment in Table 11. As noticed, our *CH2GCN* outperforms *H2GCN*.

Table 11: Classification accuracy ($\pm$ standard deviation) of the models on different benchmark node classification datasets. The higher the accuracy (in %) the better the model. ① CH2GCN with nodes centrality, ② H2GCN. Highlighted are the **best results**.

|   | Model | Texas | Cornell | Wisconsin | chameleon |
|---|---|---|---|---|---|
|  | CH2GCN w/ $\mathbf{D}$ | **79.73 (5.02)** | 68.65 (5.16) | 79.80 (4.02) | **67.89 (4.23)** |
| ① | CH2GCN w/ $\mathbf{V}_{core}$ | 78.92 (5.77) | **68.92 (7.28)** | **79.80 (3.40)** | 60.00 (5.63) |
|  | CH2GCN w/ $\mathbf{V}_{\ell\text{-walks}}$ | 78.11 (6.10) | 68.92 (6.19) | 82.35 (5.04) | 44.28 (2.32) |
|  | CH2GCN w/ $\mathbf{V}_{PR}$ | 60.27 (5.41) | 44.86 (7.76) | 52.35 (7.75) | 31.95 (5.79) |
| ② | H2GCN | 56.76 (6.73) | 51.08 (6.89) | 55.29 (5.10) | 63.93 (2.07) |

# J PROOFS OF PROPOSITIONS

In this section, we details the proofs of the propositions 3.1, 3.2 and 3.4.

### J.1 PROOF OF PROPOSITION 3.1

*Proof of Proposition 3.1.* We first prove that the operator $M_G$ is self-adjoint.
For $\varphi_1, \varphi_2 \in L^2(G)$, we have:

$$
< M_G \varphi_1, \varphi_2 >_G = \sum_{i \in \mathcal{V}} v(i) \left( M_G \varphi_1 \right)(i) \bar{\varphi}_2(i)
$$

$$
= \sum_{i \in \mathcal{V}} v(i) \left( \frac{1}{v(i)} \sum_{j \in \mathcal{N}_i} \varphi_1(j) \right) \bar{\varphi}_2(i)
$$

$$
= \sum_{i \in \mathcal{V}} \bar{\varphi}_2(i) \left( \sum_{j \in \mathcal{N}_i} \varphi_1(j) \right)
$$

$$
= \sum_{i \in \mathcal{V}} \sum_{j \in \mathcal{N}_i} a_{i,j} \bar{\varphi}_2(i) \varphi_1(j)
$$

$$
= \sum_{i,j \in \mathcal{V}} a_{i,j} \bar{\varphi}_2(i) \varphi_1(j)
$$

Similarly, we also have that,

$$
< \varphi_1, M_G \varphi_2 >_G = \sum_{i \in \mathcal{V}} v(i) \varphi_1(i) \overline{(M_G \varphi_2)}(i)
$$

$$
= \sum_{i \in \mathcal{V}} v(i) \varphi_1(i) \overline{\left( \frac{1}{v(i)} \sum_{j \in \mathcal{N}_i} \varphi_2(j) \right)}
$$

$$
= \sum_{i \in \mathcal{V}} \varphi_1(i) \overline{\left( \sum_{j \in \mathcal{N}_i} \varphi_2(j) \right)}
$$

$$
= \sum_{i,j \in \mathcal{V}} a_{i,j} \bar{\varphi}_2(i) \varphi_1(j)
$$

$$
= \sum_{i,j \in \mathcal{V}} a_{j,i} \bar{\varphi}_2(j) \varphi_1(i)
$$

Thus,

$$
\forall \varphi_1, \varphi_2 \in L^2(G), \begin{cases} < M_G \varphi_1, \varphi_2 >_G = \sum_{i,j \in \mathcal{V}} a_{i,j} \bar{\varphi}_2(i) \varphi_1(j), \\ < \varphi_1, M_G \varphi_2 >_G = \sum_{i,j \in \mathcal{V}} a_{j,i} \bar{\varphi}_2(j) \varphi_1(i). \end{cases} \tag{5}
$$

Since $a_{i,j} = a_{j,i}$, we conclude that $M_G$ is self-adjoint, i.e.

$$
< M_G \varphi_1, \varphi_2 >_G = < \varphi_1, M_G \varphi_2 >_G
$$

$M_G$ is self-adjoint, the space $L^2(G)$ is finite-dimensional, thus is diagonalizable in an orthonormal basis, and its eigenvalues are real.
We define the following norm,

$$
\|M_G\| = \sup_{\varphi \neq 0} \frac{\langle M_G \varphi, \varphi \rangle_G}{\|\varphi\|^2}.
$$

We will now prove that all eigenvalues have absolute values at most $\gamma = \min_{i \in \mathcal{V}} v(i)/deg(i)$. For that, we will first compute the two inner-products $< (I - M_G) \varphi, \varphi >_G$ and $< (I + M_G) \varphi, \varphi >_G$. For any $\varphi \in L^2(G)$, using (5), we have that:

$$\begin{cases} <\varphi,\varphi>_G = \sum_{i\in\mathcal{V}} v(i)|\varphi(i)|^2, \\ <M_G\varphi,\varphi>_G = \sum_{i,j\in\mathcal{V}} a(i,j)\bar{\varphi}(i)\varphi(j). \end{cases}$$

Let's first take the simple case, where $\gamma = \min_{i\in\mathcal{V}}\left(\frac{v(i)}{deg(i)}\right) \leq 1$, then,

$$\begin{aligned} 2<\varphi,\varphi>_G &= 2\sum_{i\in\mathcal{V}} v(i)|\varphi(i)|^2 \\ &\geq 2\gamma\sum_{i\in\mathcal{V}} deg(i)|\varphi(i)|^2 \\ &\geq 2\sum_{i\in\mathcal{V}} deg(i)|\varphi(i)|^2 \\ &\geq 2\sum_{i\in\mathcal{V}}\left(\sum_{j\in\mathcal{V}} a(i,j)\right)|\varphi(i)|^2 \\ &\geq 2\sum_{i,j\in\mathcal{V}} a(i,j)|\varphi(i)|^2 \end{aligned}$$

Therefore,

$$\begin{aligned} 2<(I-M_G)\varphi,\varphi>_G &= 2<\varphi,\varphi>_G - 2<M_G\varphi,\varphi>_G \\ &\geq 2\sum_{i,j\in\mathcal{V}} a(i,j)|\varphi(i)|^2 - 2\sum_{i,j\in\mathcal{V}} a(i,j)\bar{\varphi}(i)\varphi(j) \\ &\geq \sum_{i,j\in\mathcal{V}} a(i,j)|\varphi(i)|^2 + \sum_{i,j\in\mathcal{V}} a(i,j)|\varphi(j)|^2 - 2\sum_{i,j\in\mathcal{V}} a(i,j)\bar{\varphi}(i)\varphi(j) \\ &\geq \sum_{i,j\in\mathcal{V}} a(i,j)|\varphi(i)-\varphi(j)|^2 \end{aligned}$$

Similarly, we can prove that,

$$2<(I+M_G)\varphi,\varphi>_G \geq \sum_{i,j\in\mathcal{V}} a(i,j)|\varphi(i)+\varphi(j)|^2$$

Therefore, if $\phi \neq 0$, then,

$$\begin{cases} <(I-M_G)\varphi,\varphi>_G \geq 0, \\ <(I+M_G)\varphi,\varphi>_G \geq 0 \end{cases} \Rightarrow <\varphi,\varphi>_G \leq <M_G\varphi,\varphi>_G \leq <\varphi,\varphi>_G$$

$$\Rightarrow \frac{|<M_G\varphi,\varphi>_G|}{<\varphi,\varphi>_G} \leq 1$$

Thus, $\|M_G\| \leq 1$, i.e. all the eigenvalues have absolute values at most 1. Let now consider the general case, where $\gamma$ is not necessarily smaller than 1. Let's consider $\tilde{V} = \frac{1}{\gamma}V = diag(\frac{v(1)}{\gamma}, \ldots, \frac{v(N)}{\gamma})$. Since,

$$\tilde{\gamma} = \min_{i \in \mathcal{V}} \left( \frac{v\tilde{(i)}}{deg(i)} \right)$$

$$= \frac{1}{\gamma} \left( \frac{v(i)}{deg(i)} \right)$$

$$= \frac{\gamma}{\gamma}$$

$$= 1.$$

Therefore, all the eigenvalues of $\tilde{M}_G = \tilde{V}^{-1}A = \frac{1}{\gamma}\mathbf{V}^{-1}\mathbf{A} = \frac{1}{\gamma}M_G$ have absolute values at most 1. Thus, all the eigenvalues of $M_G$ have absolute values at most $\gamma$. $\square$

## J.2 PROOF OF PROPOSITION 3.2

*Proof of Proposition 3.2.* We will prove the first property.

We consider $P$ as the number of connected components, i.e. $G = \bigcup_{i=1}^{P} \mathcal{C}_i$.

The adjacency matrix of the graph $G$ is,

$$A = \begin{bmatrix} A_{\mathcal{C}_1} & 0 & & 0 \\ & \ddots & & \\ 0 & & A_{\mathcal{C}_i} & 0 \\ & & & \ddots \\ 0 & & 0 & A_{\mathcal{C}_P} \end{bmatrix}$$

And the transformation of $A$ by the Markov Average operator $M_G$ is,

$$M_G = \begin{bmatrix} M_{\mathcal{C}_1} & 0 & & 0 \\ & \ddots & & \\ 0 & & M_{\mathcal{C}_i} & 0 \\ & & & \ddots \\ 0 & & 0 & M_{\mathcal{C}_P} \end{bmatrix}$$

According to Proposition 3.1, for each connected component $\mathcal{C}_i$, the matrix $M_{\mathcal{C}_i}$ is diagonalizable in an orthonormal basis, and its eigenvalues are real numbers. We denote by $\mathbf{e}^{\mathcal{C}_i} = [\mathbf{e}_1^{\mathcal{C}_i}, \ldots, \mathbf{e}_{|\mathcal{C}_i|}^{\mathcal{C}_i}]$ the eigenvectors basis of $M_{\mathcal{C}_i}$ corresponding the eigenvalues $\lambda^{\mathcal{C}_i} = [\lambda_1^{\mathcal{C}_i}, \ldots, \lambda_{|\mathcal{C}_i|}^{\mathcal{C}_i}]$.

We consider the set of vectors

$$\mathbf{e} = \begin{bmatrix} \mathbf{e}^{\mathcal{C}_1} & 0 & & 0 \\ & \ddots & & \\ 0 & & \mathbf{e}^{\mathcal{C}_i} & 0 \\ & & & \ddots \\ 0 & & 0 & \mathbf{e}^{\mathcal{C}_P} \end{bmatrix}$$

The column vectors of $\mathbf{e}$ are eigenvectors of the matrix $M_G$, and which achieves the conditions of Property 1. Let's now prove the formulas of the mean and standard deviation of the $M_G$ spectrum. The matrix $\mathbf{V}^{-1}\mathbf{A}$ is defined as follow,

$$\forall 1 \leq i, j \leq n, \quad (\mathbf{D}^{-1}\mathbf{A})_{i,j} = \frac{1}{v(i)} A_{i,j}$$

Therefore, the diagonal elements of the matrix $\left(\mathbf{D}^{-1}\mathbf{A}\right)^2$ is defined as follow,

$$\forall 1 \leq i \leq n, \ \left(\left(\mathbf{D}^{-1}\mathbf{A}\right)^2\right)_{i,i} = \left(\mathbf{D}^{-1}\mathbf{A}\mathbf{D}^{-1}\mathbf{A}\right)_{i,i} \tag{6}$$

$$= \sum_j (\mathbf{D}^{-1}\mathbf{A})_{i,k}(\mathbf{D}^{-1}\mathbf{A})_{k,i} \tag{7}$$

$$= \sum_j \frac{A_{i,j} A_{j,i}}{v(i) \times v(j)} \tag{8}$$

$$= \sum_j \frac{A_{i,j}^2}{v(i) \times v(j)} \tag{9}$$

$$= \sum_{j \in \mathcal{N}_i} \frac{1}{v(i) \times v(j)} \tag{10}$$

Thus,

$$\mu\left(sp_{M_G}\right) = Mean\left(Spectrum\left[\mathbf{V}^{-1}\mathbf{A}\right]\right)$$

$$= \frac{1}{n} Sum\left(Spectrum\left[\mathbf{V}^{-1}\mathbf{A}\right]\right)$$

$$= \frac{1}{n} \sum_{i=1}^n (\mathbf{D}^{-1}\mathbf{A})_{i,i}$$

$$= \frac{1}{n} \sum_{i=1}^n \frac{1}{v(i)} A_{i,i}$$

$$= \frac{1}{n} \sum_{i=1}^n \frac{1}{v(i)},$$

and,

$$\sigma\left(sp_{M_G}\right) = Stdev\left(Spectrum\left[\mathbf{V}^{-1}\mathbf{A}\right]\right)$$

$$= \sqrt{\frac{1}{n} \sum_{\lambda \in Spectrum[\mathbf{V}^{-1}\mathbf{A}]} \left(\lambda - Mean\left(sp_\phi\right)\right)^2}$$

$$= \sqrt{\left(\frac{1}{n} \sum_{\lambda \in Spectrum[\mathbf{V}^{-1}\mathbf{A}]} \lambda^2\right) - Mean\left(sp_\phi\right)^2}$$

$$= \sqrt{\left(\frac{1}{n}Sum(Spectrum\left[\phi^2\right])\right) - Mean\left(sp_\phi\right)^2}$$

$$= \sqrt{\left(\frac{1}{n}Sum(Spectrum\left[\left(\mathbf{D}^{-1}\mathbf{A}\right)^2\right])\right) - Mean\left(sp_\phi\right)^2}$$

$$= \sqrt{\left(\frac{1}{n}Tr\left[\left(\mathbf{D}^{-1}\mathbf{A}\right)^2\right]\right) - Mean\left(sp_\phi\right)^2}$$

$$= \sqrt{\left(\frac{1}{n} \sum_{i=1} \sum_{j \in \mathcal{N}_i} \frac{1}{v(i) \times v(j)}\right) - Mean\left(sp_\phi\right)^2}$$

$$= \sqrt{\left(\frac{1}{n} \sum_{(i,j) \in E} \frac{1}{v(i) \times v(j)}\right) - Mean\left(sp_\phi\right)^2}$$

$\square$

## J.3 Proof of Proposition 3.4

*Proof of Proposition 3.4.* Let $W \subset V$, such that $|W| \leq \frac{1}{2}|\mathcal{V}|$

For $\varphi = \mathbb{1}_W - \mu_G(W)$ where $\mu_G(W) = \frac{|W|}{N_v}$ and $N_v = \sum_{i \in \mathcal{V}} v(i) = |\mathcal{V}|_v$

$$2 < (I - M_G)\varphi, \varphi >_G = 2 < \varphi, \varphi >_G -2 < M_G\varphi, \varphi >_G$$

$$\leq 2 \sum_{i \in \mathcal{V}} v(i)|\varphi(i)|^2 - 2 \sum_{i,j \in \mathcal{V}} a(i,j)\bar{\varphi}(i)\varphi(j)$$

$$\leq 2 \sum_{i \in \mathcal{V}} \beta \times deg(i)|\varphi(i)|^2 - 2 \sum_{i,j \in \mathcal{V}} a(i,j)\bar{\varphi}(i)\varphi(j)$$

$$\leq 2 \sum_{i \in \mathcal{V}} deg(i)|\varphi(i)|^2 - 2 \sum_{i,j \in \mathcal{V}} a(i,j)\bar{\varphi}(i)\varphi(j)$$

$$\leq 2 \sum_{i \in \mathcal{V}} \left(\sum_{j \in \mathcal{V}} a(i,j)\right) |\varphi(i)|^2 - 2 \sum_{i,j \in \mathcal{V}} a(i,j)\bar{\varphi}(i)\varphi(j)$$

$$\leq 2 \sum_{i,j \in \mathcal{V}} a(i,j)|\varphi(i)|^2 - 2 \sum_{i,j \in \mathcal{V}} a(i,j)\bar{\varphi}(i)\varphi(j)$$

$$\leq \sum_{i,j \in \mathcal{V}} a(i,j)|\varphi(i)|^2 + \sum_{i,j \in \mathcal{V}} a(i,j)|\varphi(j)|^2 - 2 \sum_{i,j \in \mathcal{V}} a(i,j)\bar{\varphi}(i)\varphi(j)$$

$$\leq \sum_{i,j \in \mathcal{V}} a(i,j)|\varphi(i) - \varphi(j)|^2$$

$$\leq \sum_{i,j \in \mathcal{V}} a(i,j)|\mathbb{1}_W(i) - \mathbb{1}_W(j)|^2$$

The non-zero terms in $\sum_{i,j \in \mathcal{V}} a(i,j)|\varphi(i) - \varphi(j)|^2$ are those where $i$ and $j$ are adjacent, but one of them is in $W$ and the other not.

$$< (I - M_G)\varphi, \varphi >_G \leq \frac{1}{2} \sum_{i,j \in \mathcal{V}} a(i,j)|\mathbb{1}_W(i) - \mathbb{1}_W(j)|^2$$

$$= \#\mathcal{E}(W)$$

There $\frac{1}{2}$ was removed because of the symmetry.

We also have that,

$$\frac{1}{N_v} < \mathbb{1}_W, \mathbb{1}_W >_G = \frac{1}{N_v} \sum_{i \in \mathcal{V}} v(i) = \mu_G(W),$$

and,

$$\frac{1}{N_v} < \mathbb{1}_W, \mu_G(W) >_G = \frac{1}{N_v} \sum_{i \in \mathcal{V}} v(i)\mu_G(W) = (\mu_G(W))^2,$$

Therefore,

$$\frac{1}{N_v} < \varphi, \varphi >_G = \frac{1}{N_v} < \mathbb{1}_W - \mu_G(W), \mathbb{1}_W - \mu_G(W) >_G$$

$$= \frac{1}{N_v} < \mathbb{1}_W, \mathbb{1}_W - \mu_G(W) >_G - \frac{1}{N_v} < \mu_G(W), \mathbb{1}_W - \mu_G(W) >_G$$

$$= \frac{1}{N_v} < \mathbb{1}_W, \mathbb{1}_W >_G - \frac{2}{N_v} < \mathbb{1}_W, \mu_G(W) >_G + \frac{1}{N_v} < \mu_G(W), \mu_G(W) >_G$$

$$= \mu_G(W) - 2 \left(\mu_G(W)\right)^2 + \left(\mu_G(W)\right)^2 \frac{1}{N_v} < 1, 1 >_G$$

$$= \mu_G(W) - 2 \left(\mu_G(W)\right)^2 + \left(\mu_G(W)\right)^2 \frac{1}{N_v} \sum_{i \in \mathcal{V}} v(i)$$

$$= \mu_G(W) - 2 \left(\mu_G(W)\right)^2 + \left(\mu_G(W)\right)^2 \frac{N_v}{N_v}$$

$$= \mu_G(W) - \left(\mu_G(W)\right)^2$$
$$= \mu_G(W) \left(1 - \mu_G(W)\right)$$
$$= \mu_G(W)\mu_G(W'),$$

where $W' = \mathcal{V} - W$

By definition,

$$\lambda_1(G) = \min_{\tilde{\varphi} \neq 0} \frac{< (I - M_G)\tilde{\varphi}, \tilde{\varphi} >_G}{< \tilde{\varphi}, \tilde{\varphi} >_G}$$

Therefore,

$$\lambda_1(G) \leq \frac{< (I - M_G)\varphi, \varphi >_G}{< \varphi, \varphi >_G}$$

$$\leq \frac{\#\mathcal{E}(W)}{N_v} \frac{N_v}{< \varphi, \varphi >_G}$$

$$\leq \frac{\#\mathcal{E}(W)}{N_v} \frac{N_v}{\mu_G(W)\mu_G(W')}$$

Since,

$$\frac{v_-}{v_+} \frac{|W|_v}{|\mathcal{V}|_v} \leq \mu_G(W) \leq \frac{v_-}{v_+} \frac{|W|_v}{|\mathcal{V}|_v}$$

Then,

$$N_v \mu_G(W)\mu_G(W') \geq N_v \frac{|W|_v}{|\mathcal{V}|_v} \frac{v_-}{v_+} \frac{|W'|_v}{|\mathcal{V}|_v}$$

$$\geq \frac{\sum_{i \in \mathcal{V}} v(i)}{|\mathcal{V}|_v} |W|_v \frac{v_-}{v_+} \frac{|W'|_v}{|\mathcal{V}|_v}$$

$$\geq \frac{\sum_{i \in \mathcal{V}} v(i)}{\sum_{i \in \mathcal{V}} v(i)} |W|_v \frac{v_-}{v_+} \frac{|W'|_v}{|\mathcal{V}|_v}$$

$$\geq |W|_v \frac{v_-}{v_+} \frac{|W'|_v}{|\mathcal{V}|_v}$$

$$\geq \frac{v_-}{v_+} |W|_v \frac{|W'|_v}{|\mathcal{V}|_v}$$

$$\geq \frac{v_-}{2v_+} |W|_v$$

because

$$\begin{cases} |W|_v \leq \frac{1}{2}|\mathcal{V}|_v \\ W' = \mathcal{V} - W \end{cases} \Rightarrow |W'|_v \geq \frac{1}{2}|\mathcal{V}|_v$$

Thus,

$$\forall W \subset \mathcal{V}, |W|_v \leq \frac{1}{2}|\mathcal{V}|_v \Rightarrow \lambda_1(G) \leq \frac{2v_+}{v_-}\frac{\#\mathcal{E}(W)}{|W|_v}$$

Thus,

$$\lambda_1(G) \leq \frac{2v_+}{v_-}N_v h(G) \leq 2N\frac{v_+^2}{v_-}h_v(G)$$

$\square$

## K  AVERAGE DEGREE OF A BARABASI–ALBERT MODEL

**Lemma.** Let $G^{BA}$ be a Barabasi–Albert graph of $N$ nodes generated with the hyperparameters $N_0 < N$ the initial number of nodes, $r_0 \leq N_0^2$ the initial number of random edges and $r$ the number of added edges at each time step, Then the average degree in the network is,

$$\overline{deg}(G^{BA}) = 2r + 2\frac{r_0}{N} - 2N_0\frac{r}{N}$$

and thus, as the number of nodes grows, i.e. $N \to \infty$, the average degree becomes

$$\overline{deg}(G^{BA}) \sim 2r$$

*Proof.* We start with a small graph of $N_0$ nodes and $r_0$ edges. At each time step, we increase the number of edges by $r$. Thus, if $N$ is the number of nodes at a certain time step, then there are exactly $r_0 + r(N - N_0)$ edges.

As each edge contributes to the degree of two nodes, thus, the average degree is twice the number of edges divided by the number of nodes $N$. Therefore,

$$\overline{deg}(G^{BA}) = \frac{2}{N}(r_0 + r(N - r_0))$$
$$= 2r + 2\frac{r_0}{N} - 2N_0\frac{r}{N}$$

$\square$

# L  LEARNED PARAMETERS OF DIFFERENT CENTRALITY BASED GSOS

In this section, we present some graph properties of the used dataset. We specifically present the node density, the homophily coefficient as well as the average value of different centrality metrics. We also present the $(m_1, m_2, m_3, e_1, e_2, e_3, a)$ leaned by the GNN.

## L.1  DEGREE CENTRALITY

Table 12: Graph Properties of the used datasets and the corresponding learned hyperparameters in GAGCN w/ Degree

| Dataset | Graph Properties | | | | | | Hyperparameters | | | | | | |
|---|---|---|---|---|---|---|---|---|---|---|---|---|---|
| | density | Avg. Degree | Avg. PageRank | Avg. K-core | Avg. Count. Walks | homophily | $e_1$ | $e_2$ | $e_3$ | $m_1$ | $m_2$ | $m_3$ | $a$ |
| Physics | $4.16 \times 10^{-4}$ | 14.37 | $2.89 \times 10^{-5}$ | 7.71 | 449.22 | 0.931 | 0.28 (0.01) | −0.31 (0.00) | −0.32 (0.00) | 0.34 (0.01) | 1.33 (0.01) | 0.31 (0.01) | 1.36 (0.01) |
| Photo | $4.07 \times 10^{-3}$ | 31.13 | $1.30 \times 10^{-4}$ | 16.97 | 3204.098 | 0.827 | 0.39 (0.06) | −0.26 (0.01) | −0.25 (0.01) | 0.59 (0.05) | 1.51 (0.01) | 0.53 (0.04) | 1.70 (0.04) |
| Cora | $1.43 \times 10^{-3}$ | 3.89 | $3.69 \times 10^{-4}$ | 2.31 | 42.52 | 0.809 | 0.31 (0.04) | 0.02 (0.01) | −0.02 (0.01) | 0.67 (0.02) | 1.43 (0.04) | 0.66 (0.02) | 0.69 (0.01) |
| CS | $4.87 \times 10^{-4}$ | 8.93 | $5.45 \times 10^{-5}$ | 4.94 | 162.75 | 0.808 | 0.33 (0.00) | −0.25 (0.00) | −0.26 (0.00) | 0.44 (0.01) | 1.44 (0.00) | 0.40 (0.01) | 1.47 (0.01) |
| PubMed | $2.28 \times 10^{-4}$ | 4.49 | $5.07 \times 10^{-5}$ | 2.39 | 75.43 | 0.802 | 0.28 (0.01) | −0.27 (0.00) | −0.28 (0.00) | 0.39 (0.00) | 1.40 (0.01) | 0.38 (0.00) | 1.39 (0.01) |
| Computers | $2.60 \times 10^{-3}$ | 35.75 | $7.27 \times 10^{-5}$ | 18.84 | 6221.39 | 0.777 | 0.40 (0.05) | −0.74 (0.02) | 0.24 (0.03) | 0.74 (0.05) | 1.60 (0.05) | 0.66 (0.04) | 0.86 (0.10) |
| CiteSeer | $8.22 \times 10^{-4}$ | 2.73 | $3.00 \times 10^{-4}$ | 1.73 | 18.91 | 0.735 | 0.35 (0.00) | −0.21 (0.01) | −0.22 (0.01) | 0.49 (0.01) | 1.49 (0.01) | 0.47 (0.01) | 1.50 (0.01) |
| ogbn-arxiv | $8.07 \times 10^{-5}$ | 13.67 | $5.90 \times 10^{-6}$ | 7.13 | 4898.16 | 0.654 | −0.08 (0.02) | −0.29 (0.01) | −0.41 (0.00) | 0.13 (0.01) | 1.31 (0.04) | 0.13 (0.01) | 1.00 (0.01) |
| deezer-europe | $2.31 \times 10^{-4}$ | 6.55 | $3.53 \times 10^{-5}$ | 3.57 | 106.16 | 0.525 | 0.31 (0.04) | −0.51 (0.03) | −0.54 (0.02) | 0.59 (0.04) | −0.96 (0.04) | 1.55 (0.03) | −0.59 (0.03) |
| Penn94 | $1.57 \times 10^{-3}$ | 65.56 | $2.40 \times 10^{-5}$ | 33.68 | 10662.08 | 0.470 | 0.51 (0.01) | −1.00 (0.02) | −0.09 (0.02) | 0.98 (0.01) | 1.01 (0.04) | 0.82 (0.01) | 0.95 (0.03) |
| chameleon | $1.21 \times 10^{-2}$ | 27.57 | $4.39 \times 10^{-4}$ | 16.60 | 2913.48 | 0.231 | 0.15 (0.04) | −0.06 (0.01) | −0.06 (0.01) | −0.17 (0.03) | 0.88 (0.02) | −0.16 (0.03) | −0.15 (0.02) |
| squirrel | $1.46 \times 10^{-2}$ | 76.30 | $1.92 \times 10^{-4}$ | 41.55 | 31888.02 | 0.222 | 0.38 (0.05) | −0.26 (0.03) | −0.24 (0.03) | 0.31 (0.80) | 1.75 (0.07) | 0.26 (0.70) | 1.69 (0.56) |
| arxiv-year | $8.07 \times 10^{-5}$ | 6.88 | $5.90 \times 10^{-6}$ | 7.13 | 82.85 | 0.218 | −0.25 (0.01) | −0.27 (0.01) | −0.40 (0.01) | 0.01 (0.01) | 0.99 (0.01) | 0.05 (0.01) | 0.80 (0.01) |
| Wisconsin | $1.48 \times 10^{-2}$ | 3.64 | $3.98 \times 10^{-3}$ | 2.05 | 76.26 | 0.206 | 0.95 (0.05) | −0.09 (0.04) | −0.05 (0.01) | 1.27 (0.25) | −0.94 (0.05) | 0.66 (0.07) | −0.64 (0.06) |
| Cornell | $1.68 \times 10^{-2}$ | 3.04 | $5.46 \times 10^{-3}$ | 1.74 | 58.47 | 0.132 | 0.88 (0.05) | −0.17 (0.07) | −0.07 (0.03) | 1.04 (0.29) | −0.86 (0.11) | 0.80 (0.10) | −0.78 (0.08) |
| Texas | $1.77 \times 10^{-2}$ | 3.13 | $5.46 \times 10^{-3}$ | 1.71 | 70.72 | 0.111 | 0.93 (0.03) | −0.09 (0.04) | −0.05 (0.01) | 1.17 (0.20) | −0.98 (0.05) | 0.65 (0.07) | −0.64 (0.07) |

## L.2  $k$-CORE CENTRALITY

Table 13: Graph Properties of the used datasets and the corresponding learned hyperparameters in GAGCN w/ K-Core

| Dataset | Graph Properties | | | | | | Hyperparameters | | | | | | |
|---|---|---|---|---|---|---|---|---|---|---|---|---|---|
| | density | Avg. Degree | Avg. PageRank | Avg. K-core | Avg. Count. Walks | homophily | $e_1$ | $e_2$ | $e_3$ | $m_1$ | $m_2$ | $m_3$ | $a$ |
| Physics | $4.16 \times 10^{-4}$ | 14.37 | $2.89 \times 10^{-5}$ | 7.71 | 449.22 | 0.931 | 0.38 (0.03) | −0.35 (0.01) | −0.35 (0.01) | 0.34 (0.02) | 1.28 (0.01) | 0.30 (0.02) | 1.35 (0.02) |
| Photo | $4.07 \times 10^{-3}$ | 31.13 | $1.30 \times 10^{-4}$ | 16.97 | 3204.098 | 0.827 | 0.52 (0.02) | −0.31 (0.01) | −0.31 (0.01) | 0.73 (0.03) | 1.44 (0.02) | 0.59 (0.03) | 1.77 (0.05) |
| Cora | $1.43 \times 10^{-3}$ | 3.89 | $3.69 \times 10^{-4}$ | 2.31 | 42.52 | 0.809 | 0.34 (0.01) | −0.74 (0.00) | 0.24 (0.01) | 0.60 (0.01) | 1.55 (0.01) | 0.59 (0.01) | 0.68 (0.01) |
| CS | $4.87 \times 10^{-4}$ | 8.93 | $5.45 \times 10^{-5}$ | 4.94 | 162.75 | 0.808 | 0.41 (0.01) | −0.29 (0.01) | −0.26 (0.00) | 0.43 (0.01) | 1.39 (0.01) | 0.39 (0.01) | 1.47 (0.01) |
| PubMed | $2.28 \times 10^{-4}$ | 4.49 | $5.07 \times 10^{-5}$ | 2.39 | 75.43 | 0.802 | 0.27 (0.00) | −0.31 (0.00) | −0.32 (0.00) | 0.34 (0.01) | 1.34 (0.01) | 0.34 (0.01) | 1.35 (0.01) |
| Computers | $2.60 \times 10^{-3}$ | 35.75 | $7.27 \times 10^{-5}$ | 18.84 | 6221.39 | 0.777 | 0.51 (0.02) | −0.28 (0.01) | −0.29 (0.01) | 0.78 (0.03) | 1.50 (0.01) | 0.66 (0.03) | 1.72 (0.05) |
| CiteSeer | $8.22 \times 10^{-4}$ | 2.73 | $3.00 \times 10^{-4}$ | 1.73 | 18.91 | 0.735 | 0.39 (0.01) | −0.26 (0.00) | −0.26 (0.00) | 0.45 (0.01) | 1.43 (0.01) | 0.44 (0.01) | 1.46 (0.00) |
| ogbn-arxiv | $8.07 \times 10^{-5}$ | 13.67 | $5.90 \times 10^{-6}$ | 7.13 | 4898.16 | 0.654 | 0.27 (0.01) | −0.53 (0.01) | −0.55 (0.01) | −0.70 (0.01) | −1.04 (0.03) | 0.31 (0.02) | 0.70 (0.02) |
| deezer-europe | $2.31 \times 10^{-4}$ | 6.55 | $3.53 \times 10^{-5}$ | 3.57 | 106.16 | 0.525 | −0.01 (0.03) | −0.51 (0.00) | −0.51 (0.00) | 0.02 (0.01) | 0.99 (0.01) | 0.02 (0.01) | 1.07 (0.01) |
| Penn94 | $1.57 \times 10^{-3}$ | 65.56 | $2.40 \times 10^{-5}$ | 33.68 | 10662.08 | 0.470 | −0.09 (0.30) | −0.39 (0.08) | −0.40 (0.08) | 0.28 (0.36) | 1.27 (0.16) | 0.05 (0.41) | 1.60 (0.15) |
| chameleon | $1.21 \times 10^{-2}$ | 27.57 | $4.39 \times 10^{-4}$ | 16.60 | 2913.48 | 0.231 | 0.15 (0.04) | −0.06 (0.01) | −0.06 (0.01) | −0.17 (0.02) | 0.88 (0.01) | −0.16 (0.02) | −0.15 (0.01) |
| squirrel | $1.46 \times 10^{-2}$ | 76.30 | $1.92 \times 10^{-4}$ | 41.55 | 31888.02 | 0.222 | 0.46 (0.02) | −0.78 (0.01) | 0.24 (0.01) | −0.97 (0.05) | 1.78 (0.04) | −0.97 (0.05) | −1.08 (0.12) |
| arxiv-year | $8.07 \times 10^{-5}$ | 6.88 | $5.90 \times 10^{-6}$ | 7.13 | 82.85 | 0.218 | 0.34 (0.05) | −0.36 (0.01) | −0.41 (0.01) | −0.13 (0.06) | 1.03 (0.01) | −0.03 (0.02) | 0.80 (0.02) |
| Wisconsin | $1.48 \times 10^{-2}$ | 3.64 | $3.98 \times 10^{-3}$ | 2.05 | 76.26 | 0.206 | 1.20 (0.02) | −0.05 (0.02) | −0.06 (0.02) | 1.48 (0.03) | −0.96 (0.04) | 0.54 (0.02) | −0.54 (0.03) |
| Cornell | $1.68 \times 10^{-2}$ | 03.04 | $5.46 \times 10^{-3}$ | 1.74 | 58.47 | 0.132 | 0.34 (0.05) | −0.36 (0.01) | −0.41 (0.01) | −0.13 (0.06) | 1.03 (0.01) | −0.03 (0.02) | 0.80 (0.02) |
| Texas | $1.77 \times 10^{-2}$ | 3.13 | $5.46 \times 10^{-3}$ | 1.71 | 70.72 | 0.111 | 0.50 (0.06) | −0.02 (0.02) | −0.04 (0.02) | −0.87 (0.05) | 1.04 (0.05) | −0.85 (0.05) | −0.86 (0.05) |

### L.3 PAGERANK CENTRALITY

Table 14: Graph Properties of the used datasets and the corresponding learned hyperparameters in GAGCN w/ PageRank

| Dataset | Graph Properties | | | | | | Hyperparameters | | | | | | |
|---|---|---|---|---|---|---|---|---|---|---|---|---|---|
| | density | Avg. Degree | Avg. PageRank | Avg. K-core | Avg. Count. Walks | homophily | $e_1$ | $e_2$ | $e_3$ | $m_1$ | $m_2$ | $m_3$ | $a$ |
| Physics | $4.16 \times 10^{-4}$ | 14.37 | $2.89 \times 10^{-5}$ | 7.71 | 449.22 | 0.931 | 0.51 (0.00) | 0.00 (0.00) | 0.00 (0.00) | 1.31 (0.06) | 1.00 (0.08) | 0.34 (0.06) | 0.33 (0.07) |
| Photo | $4.07 \times 10^{-3}$ | 31.13 | $1.30 \times 10^{-4}$ | 16.97 | 3204.098 | 0.827 | 0.53 (0.02) | 0.08 (0.01) | 0.08 (0.01) | 0.88 (0.01) | 0.85 (0.01) | $-0.12$ (0.01) | $-0.13$ (0.01) |
| Cora | $1.43 \times 10^{-3}$ | 3.89 | $3.69 \times 10^{-4}$ | 2.31 | 42.52 | 0.809 | 0.00 (0.00) | $-0.71$ (0.02) | 0.11 (0.01) | 0.63 (0.01) | 1.49 (0.02) | 0.63 (0.01) | 0.67 (0.01) |
| CS | $4.87 \times 10^{-4}$ | 8.93 | $5.45 \times 10^{-5}$ | 4.94 | 162.75 | 0.808 | 0.00 (0.00) | $-0.10$ (0.01) | $-0.10$ (0.01) | 0.46 (0.04) | 1.38 (0.10) | 0.46 (0.04) | 1.49 (0.03) |
| PubMed | $2.28 \times 10^{-4}$ | 4.49 | $5.07 \times 10^{-5}$ | 2.39 | 75.43 | 0.802 | 0.51 (0.00) | 0.00 (0.00) | 0.00 (0.00) | 1.34 (0.02) | 1.28 (0.02) | 0.36 (0.02) | 0.38 (0.02) |
| Computers | $2.60 \times 10^{-3}$ | 35.75 | $7.27 \times 10^{-5}$ | 18.84 | 6221.39 | 0.777 | 0.00 (0.00) | $-0.42$ (0.00) | $-0.42$ (0.00) | $-0.13$ (0.02) | 0.84 (0.01) | $-0.13$ (0.02) | 0.87 (0.02) |
| CiteSeer | $8.22 \times 10^{-4}$ | 2.73 | $3.00 \times 10^{-4}$ | 1.73 | 18.91 | 0.735 | 0.00 (0.00) | $-0.14$ (0.00) | $-0.12$ (0.00) | 0.44 (0.01) | 1.41 (0.00) | 0.44 (0.01) | 1.47 (0.01) |
| ogbn-arxiv | $8.07 \times 10^{-5}$ | 13.67 | $5.90 \times 10^{-6}$ | 7.13 | 4898.16 | 0.654 | 0.00 (0.00) | $-0.89$ (0.01) | 0.11 (0.01) | $-0.22$ (0.01) | 0.78 (0.01) | $-0.22$ (0.01) | $-0.22$ (0.01) |
| deezer-europe | $2.31 \times 10^{-4}$ | 6.55 | $3.53 \times 10^{-5}$ | 3.57 | 106.16 | 0.525 | 0.57 (0.05) | 0.05 (0.01) | 0.05 (0.01) | 0.90 (0.03) | 0.90 (0.02) | $-0.10$ (0.03) | $-0.10$ (0.02) |
| Penn94 | $1.57 \times 10^{-3}$ | 65.56 | $2.40 \times 10^{-5}$ | 33.68 | 10662.08 | 0.470 | 0.54 (0.01) | 0.05 (0.01) | 0.05 (0.01) | 1.10 (0.01) | $-0.90$ (0.01) | 0.10 (0.01) | $-0.10$ (0.01) |
| chameleon | $1.21 \times 10^{-2}$ | 27.57 | $4.39 \times 10^{-4}$ | 16.60 | 2913.48 | 0.231 | $-0.01$ (0.00) | $-0.94$ (0.00) | 0.06 (0.01) | 0.27 (0.05) | $-0.88$ (0.02) | 1.26 (0.05) | $-0.25$ (0.05) |
| squirrel | $1.46 \times 10^{-2}$ | 76.30 | $1.92 \times 10^{-4}$ | 41.55 | 31888.02 | 0.222 | 0.00 (0.00) | $-0.43$ (0.01) | $-0.43$ (0.01) | 0.14 (0.01) | $-0.86$ (0.01) | 1.14 (0.01) | $-0.14$ (0.01) |
| arxiv-year | $8.07 \times 10^{-5}$ | 6.88 | $5.90 \times 10^{-6}$ | 7.13 | 82.85 | 0.218 | 0.00 (0.00) | $-0.84$ (0.03) | 0.06 (0.01) | $-0.04$ (0.02) | 0.86 (0.03) | $-0.04$ (0.02) | $-0.05$ (0.03) |
| Wisconsin | $1.48 \times 10^{-2}$ | 3.64 | $3.98 \times 10^{-3}$ | 2.05 | 76.26 | 0.206 | 0.64 (0.02) | 0.10 (0.01) | 0.10 (0.01) | 1.68 (0.03) | $-1.01$ (0.04) | 0.69 (0.02) | $-0.69$ (0.02) |
| Cornell | $1.68 \times 10^{-2}$ | 3.04 | $5.46 \times 10^{-3}$ | 1.74 | 58.47 | 0.132 | 0.64 (0.03) | 0.11 (0.01) | 0.11 (0.01) | 1.71 (0.02) | $-1.03$ (0.05) | 0.71 (0.01) | $-0.72$ (0.01) |
| Texas | $1.77 \times 10^{-2}$ | 3.13 | $5.46 \times 10^{-3}$ | 1.71 | 70.72 | 0.111 | 0.68 (0.03) | 0.14 (0.03) | 0.14 (0.03) | 1.63 (0.04) | $-0.93$ (0.06) | 0.64 (0.04) | $-0.63$ (0.04) |

### L.4 COUNT OF WALKS CENTRALITY

Table 15: Graph Properties of the used datasets and the corresponding learned hyperparameters in GAGCN w/ Count of walks.

| Dataset | Graph Properties | | | | | | Hyperparameters | | | | | | |
|---|---|---|---|---|---|---|---|---|---|---|---|---|---|
| | density | Avg. Degree | Avg. PageRank | Avg. K-core | Avg. Count. Walks | homophily | $e_1$ | $e_2$ | $e_3$ | $m_1$ | $m_2$ | $m_3$ | $a$ |
| Physics | $4.16 \times 10^{-4}$ | 14.37 | $2.89 \times 10^{-5}$ | 7.71 | 449.22 | 0.931 | 0.95 (0.01) | $-0.02$ (0.02) | $-0.02$ (0.02) | 0.89 (0.02) | 0.96 (0.01) | $-0.10$ (0.01) | $-0.09$ (0.01) |
| Photo | $4.07 \times 10^{-3}$ | 31.13 | $1.30 \times 10^{-4}$ | 16.97 | 3204.098 | 0.827 | $-0.06$ (0.01) | $-0.07$ (0.00) | $-0.07$ (0.02) | $-0.05$ (0.01) | 0.87 (0.00) | $-0.04$ (0.02) | $-0.09$ (0.01) |
| Cora | $1.43 \times 10^{-3}$ | 3.89 | $3.69 \times 10^{-4}$ | 2.31 | 42.52 | 0.809 | 0.36 (0.02) | 0.03 (0.01) | 0.02 (0.01) | 0.68 (0.02) | 1.37 (0.09) | 0.63 (0.02) | 0.64 (0.01) |
| CS | $4.87 \times 10^{-4}$ | 8.93 | $5.45 \times 10^{-5}$ | 4.94 | 162.75 | 0.808 | 0.45 (0.02) | 0.03 (0.01) | 0.02 (0.01) | 0.62 (0.03) | 1.23 (0.04) | 0.47 (0.02) | 0.52 (0.02) |
| PubMed | $2.28 \times 10^{-4}$ | 4.49 | $5.07 \times 10^{-5}$ | 2.39 | 75.43 | 0.802 | 0.30 (0.02) | $-0.15$ (0.02) | $-0.16$ (0.02) | 0.60 (0.02) | 1.58 (0.03) | 0.56 (0.02) | 1.38 (0.04) |
| Computers | $2.60 \times 10^{-3}$ | 35.75 | $7.27 \times 10^{-5}$ | 18.84 | 6221.39 | 0.777 | $-0.05$ (0.01) | $-0.07$ (0.00) | $-0.07$ (0.00) | $-0.05$ (0.00) | 0.86 (0.01) | $-0.04$ (0.03) | $-0.10$ (0.02) |
| CiteSeer | $8.22 \times 10^{-4}$ | 2.73 | $3.00 \times 10^{-4}$ | 1.73 | 18.91 | 0.735 | 0.25 (0.01) | $-0.13$ (0.01) | $-0.14$ (0.01) | 0.67 (0.01) | 1.63 (0.01) | 0.62 (0.01) | 1.63 (0.01) |
| ogbn-arxiv | $8.07 \times 10^{-5}$ | 13.67 | $5.90 \times 10^{-6}$ | 7.13 | 4898.16 | 0.654 | 0.12 (0.00) | $-0.08$ (0.00) | $-0.19$ (0.01) | 0.32 (0.00) | 1.57 (0.02) | 0.32 (0.00) | 0.93 (0.01) |
| deezer-europe | $2.31 \times 10^{-4}$ | 6.55 | $3.53 \times 10^{-5}$ | 3.57 | 106.16 | 0.525 | 0.26 (0.03) | $-1.04$ (0.03) | $-0.07$ (0.03) | 0.58 (0.04) | 0.88 (0.06) | 0.60 (0.04) | 0.67 (0.06) |
| Penn94 | $1.57 \times 10^{-3}$ | 65.56 | $2.40 \times 10^{-5}$ | 33.68 | 10662.08 | 0.470 | 0.30 (0.00) | $-0.77$ (0.03) | 0.06 (0.09) | 0.58 (0.00) | $-0.46$ (0.16) | 1.39 (0.01) | $-0.31$ (0.17) |
| chameleon | $1.21 \times 10^{-2}$ | 27.57 | $4.39 \times 10^{-4}$ | 16.60 | 2913.48 | 0.231 | 0.32 (0.06) | $-0.05$ (0.01) | $-0.05$ (0.01) | $-0.28$ (0.08) | 0.89 (0.02) | $-0.17$ (0.03) | $-0.14$ (0.02) |
| squirrel | $1.46 \times 10^{-2}$ | 76.30 | $1.92 \times 10^{-4}$ | 41.55 | 31888.02 | 0.222 | 0.23 (0.11) | $-0.71$ (0.10) | 0.35 (0.10) | $-0.46$ (0.46) | 1.28 (0.22) | $-0.32$ (0.33) | $-0.29$ (0.48) |
| arxiv-year | $8.07 \times 10^{-5}$ | 6.88 | $5.90 \times 10^{-6}$ | 7.13 | 82.85 | 0.218 | 0.23 (0.01) | $-0.28$ (0.01) | $-0.16$ (0.01) | $-0.26$ (0.01) | 0.99 (0.03) | $-0.01$ (0.04) | 0.84 (0.03) |
| Wisconsin | $1.48 \times 10^{-2}$ | 3.64 | $3.98 \times 10^{-3}$ | 2.05 | 76.26 | 0.206 | 0.40 (0.01) | $-0.79$ (0.07) | 0.18 (0.05) | 0.61 (0.01) | $-1.38$ (0.08) | 1.48 (0.01) | $-0.69$ (0.06) |
| Cornell | $1.68 \times 10^{-2}$ | 03.04 | $5.46 \times 10^{-3}$ | 1.74 | 58.47 | 0.132 | 0.40 (0.01) | $-0.21$ (0.04) | $-0.22$ (0.04) | 0.59 (0.01) | $-1.51$ (0.06) | 1.47 (0.01) | $-0.61$ (0.05) |
| Texas | $1.77 \times 10^{-2}$ | 3.13 | $5.46 \times 10^{-3}$ | 1.71 | 70.72 | 0.111 | 0.40 (0.01) | $-0.72$ (0.03) | 0.24 (0.03) | 0.58 (0.01) | $-1.48$ (0.05) | 1.47 (0.01) | $-0.59$ (0.03) |

