# OpenReview forum: "Centrality Graph Shift Operators for Graph Neural Networks"
_ICLR.cc/2025/Conference — ICLR 2025 Conference Withdrawn Submission_

### Official Review · Reviewer_DvSK · 2024-10-28

**Soundness:** 2
**Presentation:** 3
**Contribution:** 2
**Rating:** 3
**Confidence:** 5

**Summary:**

This paper introduces a new approach to constructing GSOs by normalizing adjacency matrices with global centrality metrics, named Centrality GSOs (CGSOs). These operators leverage metrics like PageRank, k-core numbers, and fixed-length path counts to capture broader structural information within graphs. The authors analyze the spectral properties of CGSOs to understand their influence on graph signals and validate this understanding through spectral clustering experiments on synthetic and real-world datasets. Additionally, they demonstrate how CGSOs can be used as message-passing operators in GNNs, with improved performance in modified GCN and GAT across several benchmark datasets.

**Strengths:**

1. The proposed Centrality GSOs integrate both local and global structural information, showing promise for clustering and classification tasks.
2. The authors conducted a comprehensive theoretical and empirical analysis of the CGSOs, demonstrating their efficiency.
3. The paper is well-written, clear, and easy to follow.

**Weaknesses:**

1. The novelty of this paper is limited. The proposed CGSOs incorporate some established metrics, but their performance is comparable with that of some basic methods, such as GCN.
2. The cost of CGSOs is excessively high for large graphs due to their quadratic complexity in computing the PageRank score.
3. The CGSOs contain numerous scalar parameters that require training, which presents a significant challenge in achieving optimal results.
4. Notably, another category of polynomial-based GNNs, such as GPRGNN[1] and BernNet[2], can learn the GSO directly and demonstrate superior performance in real-world tasks.
5. The baselines used in the experiments are outdated. For the spectral clustering task, recent methods should be included for comparison. Similarly, for the classification task, newer methods like GCNII[3], GPRGNN[1], and UniFilter[4] should also be considered.

[1] Chien, Eli, et al. "Adaptive universal generalized pagerank graph neural network." In ICLR, 2021.
[2] He, Mingguo, Zhewei Wei, and Hongteng Xu. "Bernnet: Learning arbitrary graph spectral filters via bernstein approximation." In NeurIPS, 2021.
[3] Chen, Ming, et al. "Simple and deep graph convolutional networks." In ICML, 2020.
[4] Huang, Keke, Yu Guang Wang, and Ming Li. "How Universal Polynomial Bases Enhance Spectral Graph Neural Networks: Heterophily, Over-smoothing, and Over-squashing." In ICML, 2024.

**Questions:**

1. Please begin by clarifying the weaknesses section.
2. In the classification task, does the splitting of Cora, CiteSeer, and PubMed match that in the original GCN paper? If so, why is the GCN accuracy lower here than reported in the original paper?
3. Can CGSOs be applied to polynomial-based GNNs? If so, how does their performance compare? Theoretically, polynomial-based GNNs should be able to learn an arbitrary GSO.

---

### Official Review · Reviewer_hmaR · 2024-11-01

**Soundness:** 2
**Presentation:** 2
**Contribution:** 2
**Rating:** 3
**Confidence:** 4

**Summary:**

This paper proposes to construct Graph Shift Operators (GSOs) which normalize adjacency matrices by global centrality metrics such as the PageRank, $k$-core, or count of fixed length walks. The proposed Centrality GSOs (CGSOs) can introduce global information without increasing the connectivity of computational graphs of message-passing over the observed graphs.
The authors study the spectral property of CGSOs via spectral clustering on Stochastic Block Barabási–Albert Models (SBBAM) and compare them with other GNNs on several real-world datasets.

**Strengths:**

1. This paper is well written in the introduction, preliminary knowledge and related work, especially the throughout introduction on the background of GSOs and centrality.
2. The paper has a good analysis of CGSOs via spectral clustering.

**Weaknesses:**

The benefits of CGSOs have not been sufficiently verified by experiments.

1. This work "considers a learnable parameterized CGSO framework which is a generalization of the work of Dasoulas et al. (2021)". However, there is no comparison to PGSO (Dasoulas et al., 2021).
2. Besides, there is no comparison to other "parameterized GSO" works. There are several works that do not identify themselves as parameterized GSO but actually match the *Definition 2.1*, such as Directional Graph Networks (Beani et al., 2021), ACM (Luan et al., 2022), PEGN (Wang et al., 2022) and the 1-hop variant of CKGConv (Ma et al., 2024).
3. No demonstration of retaining the original connectivity is crucial, which is the main advantage of GSOs. A simple study on memory or runtime is expected for CGSOs compared to PPNP and APPNP (Gasteiger et al., 2019). What is the computation and performance trade-off?


Based on the aforementioned, I am not fully convinced that introducing global centrality is the key point of the improved performance. Alternatively, it is natural to suspect that the improvement is from parameterized GSO, allowing high-pass filtering, rather than the global centrality, since
- 5/8 datasets are heterophily graph datasets;
- no significant outperformance on the homophily graphs, CiteSeer and PubMed, compared to *GCN w/ $\hat{\mathbf{A}}$*.






----------
- Beani, D., Passaro, S., Létourneau, V., Hamilton, W., Corso, G., & Lió, P. (2021). Directional Graph Networks. _Proc. Int. Conf. Mach. Learn._, 748–758.
- Luan, S., Hua, C., Lu, Q., Zhu, J., Zhao, M., Zhang, S., Chang, X.-W., & Precup, D. (2022). Revisiting Heterophily For Graph Neural Networks. _Adv. Neural Inf. Process. Syst._, _35_, 1362–1375.
- Wang, H., Yin, H., Zhang, M., & Li, P. (2022). Equivariant and Stable Positional Encoding for More Powerful Graph Neural Networks. _Proc. Int. Conf. Learn. Represent._ International Conference on Learning Representations.
- Ma, L., Pal, S., Zhang, Y., Zhou, J., Zhang, Y., & Coates, M. (2024). CKGConv: General Graph Convolution with Continuous Kernels. _Proc. Int. Conf. Mach. Learn._

**Questions:**

Listed in the weakness

---

### Official Review · Reviewer_v7sc · 2024-11-03

**Soundness:** 2
**Presentation:** 2
**Contribution:** 2
**Rating:** 3
**Confidence:** 4

**Summary:**

The present paper studies the role of that normalization plays for graph shift operators. Many commonly utilised graph shift operators use a node-degree-based normalization. In this work the authors note tha node-degree is only one specific measure of node centrality. As a main point of the paper, it is proposed to consider also other normalisations for graph shift operators, which are based on other measures of centrality, such as PageRank, k-count and counts of fixed length walks.
A spectral analysis of the Markov averaging operator associated to a given normalization is conducted. This operator generalised the transition matrix for simple random walk on the graph at hand bz replacing the (one-sided) degree normalization with normalization by the given considered centrality metric. Subsequently, Centrality Graph Neural Networks (CGNNs) are proposed. Here it is no longer the Markov averaging operator, but a more general form of (still local) shift operator on which networks are based. The performance of GGCN and GAT(v2) models based on this operator (defined in eq. (4)) is then experimentally investigated via the task of node classification. Additionally, spectral clustering experiments based on the considered graph shift operators are performed.

**Strengths:**

The paper considers an interesting problem: The role of normalisation is indeed an understudied question for graph neural networks (GNNs).

Table 1 shows that choosing the right normalisation in a GCN can indeed positively influence performance.

The discussion of the relationship between centrality metrics, exponents and clustering results in Section 4 is interesting and sheds some light onto the inter-dependence of these quantities.

**Weaknesses:**

Unfortunately I see several weaknesses in the paper as it currently is:

To me the most important point is the disconnect between the theoretically studied Markov averaging operator (line 192) and the experimentally investigated CGSO (eq. (4) / line 285). As far as I can tell, the theoretical results only apply to the CGSOs for which $m_1, e_3, m_3 = 0$. This severely limits the applicability of the theoretical results.

It also seems that a more inclusive theoretical characterisation of such CGSOs is not out of reach: For the degree-normalization, this was e.g. achieved in https://arxiv.org/abs/2101.10050 (c.f. ibid. Section 4).

It should also be stated clearly in the present paper, that parametrising the graph shift operator as in eq. 4 is not novel: It arises from Ideas in  https://arxiv.org/abs/2101.10050 by replacing the degree normalization with a more general centrality normalization (otherwise both papers use a similar labelling of parameters, leading me to believe that the authors are aware of the earlier work).

The claim that the layer update rule specified in eq.s (1) and (2) encompasses the entirety of message passing models is not correct, as the message function considered here e.g. does not depend on the respective node features. While it is of course valid to restrict an analysis to a certain model subclass, this needs to be clearly stated. Also, I am unsure how (2) refers to the update step; to me this equation describes channel mixing. Could the authors clarify?

As for the theoretical analysis, I am unfortunately unsure about both validity and significance of claims:
In Proposition 3.1, all results except the upper bound on the spectral radius are straight forward results from linear algebra.
In Proposition 3.2, I am unsure whether the given expression of the mean of the eigenvalues is correct: The given Markov averaging operator has all diagonal entries zero. Hence the trace (computed in the weighted $L^2$ space of Section 3.1) is zero. Hence the sum and thus also the mean of the eigenvalues is zero. Are self loops considered here? Also the subscript $\phi$ in this theorem is undefined as far as I can tell.


In the experimental section, it would be good to highlight the performance gain that choosing a different normalization (i.e. centrality metric) and parametrisation of the GCSO can bring over the base model.

Also, in the paragraph on baselines, the GSO for GCN is included without self-loops. This is not the standard matrix used in GCN. Is this merely a typo?

While it is interesting to see the performance gain for GCN and GAT(v2) these methods, while standard) are fairly old by now (three to eight years).  It would be good to compare with at least some more recent or SOTA methods.

Additionally, it seems straightforward to combine the GCSO with at least one spectral method, such as e.g. ChebNet or BernNet, as these methods rely heavily on graph shift operators.

In the introduction section, I believe the historical remarks on matrix representations of graphs are superfluous. To me, this would be interesting trivia in a textbook setting. In a paper setting, I believe these column inches are better spent elsewhere.


While I like the premise of the paper, the weaknesses of the paper currently overweigh in my estimation

**Questions:**

What is the significance of knowing the mean and standard deviation of the set of eigenvalues of a short operator?

Why is the mean in Theorem 3.2 not zero?

What are the implications of Theorem 3.4?

What about spectral methods? These methods are strongly based on Graph Shift operators. Have the authors conducted experiments beyond first order methods, and e.g. considered combining the newly introduced GCSOs with (e.g.) polynomial spectral methods? If not, would it be possible to include this in an updated version?

Which matrix is used in GCN? Renormalised with self loops or not?

---

### Official Review · Reviewer_2h1E · 2024-11-04

**Soundness:** 3
**Presentation:** 3
**Contribution:** 3
**Rating:** 6
**Confidence:** 4

**Summary:**

This paper introduces the concept of Centrality Graph Shift Operators (CGSOs) and evaluates their effectiveness when integrated into Graph Neural Networks (GNNs). The authors examine three global CGSOs—k-core, PageRank, and walk-count—and provide a theoretical analysis of their spectral properties. They test the power of these CGSOs in the clustering of synthetic graphs and the Cora graph dataset.
They then incorporate these CGSOs into common GNN architectures, such as GCN and GATv2, and test their performance on several graph datasets. The results indicate that using CGSOs enhances the accuracy of the model without a significant increase in computational cost, showing the benefit of including global centrality metrics in GNN message passing.

**Strengths:**

1. **Clear and Well-Written**: The paper is well-structured and easy to follow, making the complex concepts around CGSOs accessible.

2. **Effective Method and Thorough Experiments**: The proposed CGSOs show promising results in clustering and improve GNN performance by leveraging global centrality metrics such as k-core, PageRank, and walk-count. The method is shown to enhance accuracy with low computational overhead during training, although the pre-calculations may take some time. This makes it practical for real-world applications. The authors do experiments on a variety of datasets and compare different CGSOs.

3. **Rigorous Theory**: The theoretical analysis is rigorous. The authors back up their approach with spectral analysis and proofs, adding credibility to the method.

4. **Thorough Hyperparameter Search and Insightful Results**: Conducting a hyperparameter search on the parameters of the CGSOs is indeed insightful, and the advantage of the negative loops is an interesting fact that was previously unknown to the community.

**Weaknesses:**

1. **Limited Comparisons with Strong Baselines**: While the paper demonstrates the effectiveness of CGSOs within traditional GNN models (such as GCN and GATv2), it lacks comparison with more recent and competitive methods, like Graph Transformers and other strong GNN baselines. Including such comparisons could have provided a clearer perspective on the advantages and limitations of CGSOs relative to state-of-the-art models, particularly those that also incorporate global graph structure.

2. **Inconsistent Performance Across CGSOs**: The performance of the models varies significantly depending on the CGSO used, with no single centrality metric consistently outperforming the others across datasets. This inconsistency suggests that the choice of CGSO might be heavily dataset-dependent, potentially complicating its practical application, as users would need to test different CGSOs to identify the most suitable one for their specific data. Many times also a simple degree Graph Shift Operation outperforms many of the more complicated CGSOs proposed in this work. This could limit the overall generalizability and efficiency of the proposed approach.

3. **Many Hyperparameters**: The proposed method introduces many hyperparameters, specifically those introduced in Equation 4, making hyperparameter optimization challenging.

4. **Potential Bias in Evaluation**: The authors have used two-layer GNNs for all comparisons. This limits the GNNs' power to local information, while CGSOs provide more global information even for shallow GNNs.

----
Minor point: Line 471: Singless should be corrected to Signless.

**Questions:**

1. **CGATv2 Details**: How do the CGSOs combine with CGATv2? Are the attention scores calculated first, followed by the combination, or does normalization occur after combining the CGSOs? Providing a little more detail about the implementation might be helpful.

2. **Comparison with Positional and Structural Encodings**: How does the proposed method compare to positional and structural encodings based on techniques like Laplacian and random walks? Some examples of these encodings have been used in the GraphGPS paper [1]. These encodings also avoid adding much complexity during training, although preprocessing may be time-intensive, similar to CGSOs. Given this, they seem to be a fair comparison to the methods presented in this work.

3. **Out-of-Memory Issue on Deezer-Europe**: The Deezer-Europe dataset appears to be smaller than larger datasets like arxiv-year. Why do the results show an out-of-memory (OOM) issue on Deezer-Europe but not on the larger datasets? What device were the experiments conducted on, and how much memory does the device have?

[1] Rampášek, Ladislav, et al. "Recipe for a general, powerful, scalable graph transformer." Advances in Neural Information Processing Systems 35 (2022): 14501-14515.

---

### Note · Authors · 2024-12-04

**Comment:**

We would very much like to thank the reviewers for their insightful reviews and the AC for taking the time to handle our paper. Unfortunately, we were unable to produce a comprehensive, convincing rebuttal during the discussion period and would therefore like to withdraw the paper to not take anymore of your time. We will certainly take the valuable feedback provided into account in future versions of this work. Thank you very much again.

**Withdrawal Confirmation:**

I have read and agree with the venue's withdrawal policy on behalf of myself and my co-authors.